# Accelerating Denoising Generative Models is as Easy as Predicting Second-Order Difference

## Abstract

High-fidelity diffusion and flow models remain latency-bound at inference, motivating acceleration that leaves pretrained weights untouched. We ask: what is the *minimal yet principled* way to accelerate sampling? Under a simple and mild budget, when uniform reduction targets more than $2\times$ speedup, each three-step window contains at most one fresh denoiser call, creating a structural scarcity of real signals. From this constraint, we isolate the *observed* information at step $t$—the fresh output from the denosing model $\psi_t$ and its backward difference $\Delta\psi_t^{(1)} = \psi_t - \psi_{t+1}$—and show it induces a uniquely minimal, affine-exact second-order predictor $\hat{\psi}_{t-1} = 2\psi_t - \psi_{t+1}$. We prove that, under this scarcity, the two-point second-order rule is the information-consistent optimum: it is BLUE among linear two-point estimators. Naively chaining this predictor across consecutive steps destabilizes sampling by compounding approximation errors. We resolve this by *reusing the observed tuple* in an interleaved zig–zag schedule that prevents back-to-back extrapolations and controls variance. The resulting method, **ZEUS**, is a zero-overhead, backbone- and parameterization-agnostic plug-in requiring no retraining, no feature caches, and no architectural changes. Across images and video, ZEUS consistently moves the speed–fidelity Pareto frontier outward versus recent state-of-the-art, delivering up to $3.2\times$ end-to-end speedup while improving perceptual similarity.

## 1 Introduction

Recent advances in denoising generative models (Diffusion/Flow) have set new benchmarks across image, video, text, and audio generation Sohl-Dickstein et al. (2015); Song et al. (a), substantially lowering creative barriers. Sampling in these models can be cast as transport along a reverse probability-flow ODE Song & Ermon (2019); Song et al. (b); Lipman et al. (2023); Liu et al. (2022), often requiring hundreds to thousands of steps. Modern high-order numerical solvers (Karras et al., 2022; Lu et al., 2022a;b) dramatically reduce the number of evaluations, yet wall-clock latency remains substantial when models and resolutions scale Rombach et al. (2022); Podell et al.; Chen et al. (2024a;b); Esser et al. (2024). This motivates training-free acceleration strategies, which significantly reduce inference cost without modifying pretrained weights.

Training-free strategies Yuan et al. (2024); Zhang et al. (2025a); Xi et al. (2025); Yang et al. (2025a) exploit redundancy empirically observed along the sampling trajectory of pretrained backbones Ronneberger et al. (2015); Peebles & Xie (2023). Step-wise approaches (e.g., feature caching Ma et al. (2024); Zhao et al. (2024); Wimbauer et al. (2024); Liu et al. (2024); Chen et al. (2024c); Shen et al.) reduce or bypass computation at selected steps and approximate the corresponding output from cached information, offering large acceleration granularity and easy deployment on new backbones. Recent work Yu et al. (2025); Liu et al. (2025) pushes toward higher-order extrapolation with complex approximation strategies and longer chains of consecutive reduced steps. While achieving unprecedentedly small overhead, these designs exhibit faithfulness gaps (LPIPS Zhang et al. (2018), FID Heusel et al. (2017)) relative to the original sampler. In this work, we make a counterintuitive but principled approach:

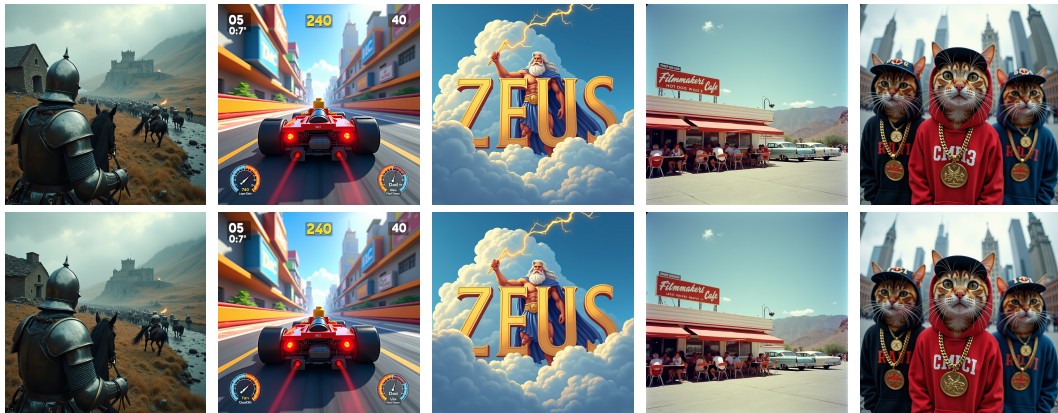

Figure 1: Accelerating `Flux.1-Dev` by $2.09\times$ with ZEUS (medium) with 50 inference steps.

***Less is more—accelerating diffusion/flow sampling is best achieved by reusing a second-order predictor derived from the observed information set.***

Consider accelerating denoising with a step-wise method that applies a uniform reduction after a single fresh computation and then approximates subsequent steps. We formalize our insight under a mild pigeonhole budget assumption: *once the acceleration ratio exceeds $2\times$ with a uniform reduction rule, two consecutive fresh evaluations cannot occur.* This induces a structural scarcity of fresh steps along the sampling trajectory. Consequently, higher-order predictors that extend their input window across multiple consecutively reduced steps draw on an increasing fraction of approximated signals whose errors accumulate across sampling, leading to faithfulness degradation. This motivates a precise question for any state $t$: *which signals are truly observed by the sampler at that moment, and how can we fully leverage them under the budget constraint?*

The freshly evaluated model output at the current state, $\psi_t$, is an observed signal. We also observed that the backward first difference $\Delta_t^{(1)} := \psi_t - \hat{\psi}_{t+1}$ is an observed, path-dependent signal that encodes the model's realized response between the received signal at $t+1$ and the fresh computation at $t$. Taken together, this *observed information set* $\{\psi_t, \Delta_t^{(1)}\}$ naturally follows by a uniquely minimal second-order predictor for the next reduced step: $\hat{\psi}_{t-1} = 2\psi_t - \hat{\psi}_{t+1}$. Ablations show that the second-order predictor yields better sampling quality and lower reconstruction error. We thus state our first observation:

***Observation 1: Under uniform reduction and $>2\times$ acceleration, second-order prediction from the observed information set is the minimal yet effective scheme for training-free acceleration.***

After determining a second-order predictor, we confront the stability challenge that arises when approximating across multiple consecutively reduced steps. Higher-order—and especially second-order—predictors are precise but can overshoot without frequent re-anchoring. In comparison, reuse-only schemes remain numerically stable yet sacrifice precision. This tension poses a natural question: *can we be both precise and stable at ambitious skipping rates?* Our answer is deliberately simple: **reuse the observed information pair** and arrange it in an interleaved schedule that never extrapolates twice in a row. When reducing computation over consecutive steps, duplicating the pair in a zig-zag pattern preserves stability—each approximate step is immediately re-anchored—while fully leveraging the available real signals to recover precision. Ablations show that this tuple-reuse strategy yields superior sampling quality and robustness under aggressive acceleration compared to reuse-only and predictor-only baselines. We thus state our second observation:

***Observation 2: Reusing the observed information pair across multiple consecutive steps achieves the precision of second order without drift.***

We call this simple, lightning–fast zig-zag rule **ZEUS**—*Zero-cost Extrapolation-based Unified Sparsity*. In the sections that follow, we present theoretical guarantees and extensive experiments showing that ZEUS is plug-and-play across diverse backbones, prediction objectives, and sampling

schedules. Despite its minimalism, ZEUS consistently improves the speed–fidelity Pareto frontier and outperforms prior training-free approaches on comprehensive evaluation suites.

Our main contributions are fourfold, naturally forming **ZEUS**:

- **Z**ero-cost: A training-free method with no finetuning, no architectural changes, and negligible runtime overhead.

- **E**xtrapolation-based: A principled second-order predictor that fully exploits observed signals under uniform skipping.

- **U**nified: A backbone- and head-agnostic framework compatible with diverse architectures, schedulers, and modalities.

- **S**parsity: A structured zig-zag reuse rule that yields stable acceleration with lower perceptual error at aggressive skip rates.

## 2 RELATED WORK

**Denoising Generative Models.** Sohl-Dickstein et al. (2015); Ho et al. (2020); Song et al. (b); Nichol & Dhariwal (2021) construct a forward perturbation of data into noise and learn a denoising network that enables sampling by integrating an associated reverse dynamics. Diffusion Ho et al. (2020); Nichol & Dhariwal (2021) and flow models Liu et al. (2022); Albergo & Vanden-Eijnden (2022); Lipman et al. (2023) have emerged as widely used and scalable frameworks for generative modeling. Recent efforts train large models with transformer-based backbones Peebles & Xie (2023) to produce high-fidelity samples across image, video, text, and audio modalities.

Sampling with denoising generative models can be cast as transport along a reverse probability-flow ODE Song & Ermon (2019); Song et al. (a), with parameterizations into various prediction objectives Song et al. (b); Song & Ermon (2019); Song et al. (a; 2023); Kim et al.; Lipman et al. (2023). Numerical ODE solvers Lu et al. (2022a;b); Karras et al. (2022) substantially reduce the number of model evaluations needed for high-quality samples. Complementary frameworks build on this probability-flow view: consistency models Song et al. (2023); Lu & Song (2024) provide a direct mapping between clean data and any point on the trajectory, while MeanFlow Geng et al. (2025) predicts an induced field of the velocity field to enable one-step generation. Despite these advances, generating from a pre-trained architecture is still time-consuming, motivating the study of training-free acceleration techniques that reduce computation without modifying weights.

Table 1: Unified network parameterizations.

| Prediction mode | Target $\psi_0(\mathbf{x}_0, \epsilon, s)$ | Reconstruction $\hat{\mathbf{x}}_0^{(s)}$ |
|---|---|---|
| $\epsilon$-prediction | $\epsilon$ | $\hat{\mathbf{x}}_0^{(s)} = \frac{1}{\alpha_s}\mathbf{x}_s - \frac{\sigma_s}{\alpha_s}\psi_\theta(\mathbf{x}_s, s)$ |
| $x_0$-prediction | $\mathbf{x}_0$ | $\hat{\mathbf{x}}_0^{(s)} = \psi_\theta(\mathbf{x}_s, s)$ |
| $v$-prediction | $v = \alpha_s\epsilon - \sigma_s\mathbf{x}_0$ | $\hat{\mathbf{x}}_0^{(s)} = \frac{\alpha_s}{\alpha_s^2+\sigma_s^2}\mathbf{x}_s - \frac{\sigma_s}{\alpha_s^2+\sigma_s^2}\psi_\theta(\mathbf{x}_s, s)$ |
| $s$-prediction | $\nabla_{\mathbf{x}_s}\log q_s(\mathbf{x}_s)$ | $\hat{\mathbf{x}}_0^{(s)} = \frac{1}{\alpha_s}\mathbf{x}_s + \frac{\sigma_s^2}{\alpha_s}\psi_\theta(\mathbf{x}_s, s)$ |
| Flow matching | $\epsilon - \mathbf{x}_0$ | $\hat{\mathbf{x}}_0^{(s)} = \mathbf{x}_s - s\,\psi_\theta(\mathbf{x}_s, s)$ |

**Training-free Acceleration.** of denoising generative models optimize various granularity levels of the denoising process. Token reduction strategies Bolya & Hoffman (2023); Kim et al. (2024) exploit spatial redundancy in image tokens. The ToCa series Zou et al. (2024); Zhang et al. (2024) combines adaptive token pruning with feature caching. Attention-focused methods include the Sparse VideoGen series Xi et al. (2025); Yang et al. (2025a), which introduces sparse attention in spatial-temporal dimension (SVG) and semantic-aware permutation (SVG2) to select and densify critical tokens for efficient GPU execution. DiTFastAttn Yuan et al. (2024); Zhang et al. (2025a) compresses the attention module according to the redundancies identified after a light search. In a higher granularity level, feature-caching methods cut sampling latency by reusing intermediate states across steps Ma et al. (2024); Zhao et al. (2024); Wimbauer et al. (2024); Liu et al. (2024); Chen et al. (2024c); Shen et al. . Recent approaches pursue higher-order prediction of features or outputs

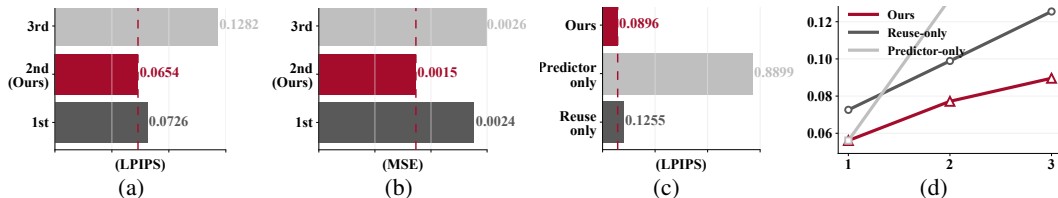

Figure 3: **Ablations of ZEUS on SDXL with DPM-Solver++ (50 steps). (a,b) Effectiveness of the second-order predictor:** (a) image quality vs. predictor order (LPIPS ↓); (b) per-step reconstruction error vs. predictor order (MSE ↓). **(c,d) Stability of reusing the observed information set:** (c) image quality at full:reduced = 1:3 under three approximation schemes (LPIPS ↓); (d) image quality vs. the number of consecutive reduced steps under the same three schemes (LPIPS ↓).

to push training-free limits. TaylorSeer Liu et al. (2025) forecasts future features from past timesteps using Taylor expansion instead of cache-then-reuse. AB-Cache Yu et al. (2025) models and predict adjacent denoising step relations with Adams–Bashforth. AdaptiveDiffusion Ye et al. (2024) and SADA Jiang et al. (2025) allocate computation dynamically, with the latter addressing step-wise and token-wise sparsity during denoising in an Adams–Moulton solver manner.

## 3 BACKGROUND

A central tool in modern denoising generative models is the *probability flow ODE (PF–ODE)*. Starting from the linear forward process $\mathbf{x}_s = \alpha_s \mathbf{x}_0 + \sigma_s \boldsymbol{\epsilon}, \ s \in [0,1]$, with data sample $\mathbf{x}_0$ and Gaussian noise $\boldsymbol{\epsilon} \sim \mathcal{N}(0, I)$, Previous works (Song & Ermon, 2019; Song et al., b) showed that the marginal laws $\{q_s\}$ can be generated not only by a reverse SDE but also by the deterministic ODE

$$d\mathbf{x}_s = \left[ f(s)\,\mathbf{x}_s - \tfrac{1}{2}g(s)^2 \nabla_{\mathbf{x}_s} \log q_s(\mathbf{x}_s) \right] ds \tag{1}$$

where $f(s)$ and $g(s)$ denote the drift and diffusion schedules determined by the forward process. The PF-ODE exactly preserves the diffusion marginals while avoiding stochasticity. Consequently, a trained network that estimates $\nabla_{\mathbf{x}_s} \log q_s(\mathbf{x}_s)$, or an equivalent target, enables fully deterministic generative sampling.

To unify different training conventions, we adopt a single notation $\psi_\theta : \mathbb{R}^d \times [0,1] \to \mathbb{R}^d$, trained to predict $\psi_0(\mathbf{x}_0, \boldsymbol{\epsilon}, s)$, where $\psi_0$ is chosen from several linearly related parameterizations. This abstraction covers $\epsilon$-prediction, $x_0$-prediction, $v$-prediction, score-prediction, and flow-matching within the same framework, enabling uniform analysis of objectives and inference rules.

## 4 ZERO-COST EXTRAPOLATION-BASED UNIFIED SPARSITY

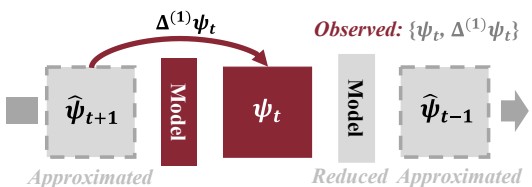

Figure 2: **Scarcity of fresh computation.** Under limited denoiser calls, the executed trajectory yields the *observed, path-wise* information set $\{\psi_t, \Delta^{(1)}\psi_t\}$, where $\Delta^{(1)}\psi_t = \psi_t - \hat{\psi}_{t+1}$.

We now develop our methods from the stepwise acceleration strategy. We bypass denoising model evaluations with a sparsity ratio $r$ and then approximate the model output $\psi$ parameterized by common prediction objectives. To formulate our acceleration paradigm during discretized ODE sampling, we pose two guiding questions that lead to our key design choices: **(i)** What is the *minimal* but *effective* approximation scheme? **(ii)** How can we approximate multiple consecutive steps without destabilizing sampling?

### 4.1 WHAT IS THE MINIMAL BUT EFFECTIVE APPROXIMATION SCHEME?

Consider a denoising generative model sampling across discrete times $\cdots > t+1 > t > t-1 > \cdots$. Training-free acceleration reduces evaluations by exploiting either step-level or feature-level reuse,

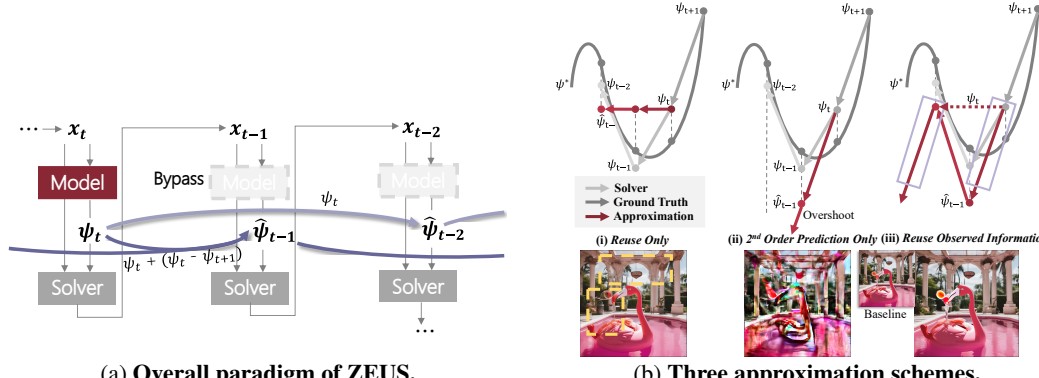

(a) **Overall paradigm of ZEUS.**   (b) **Three approximation schemes.**

Figure 4: **ZEUS overview and approximation strategies.** Panel (a) shows the pipeline; panel (b) compares reuse-only, predictor-only, and the reuse of the observed information pair. Dark gray: reference trajectory $\psi^\star$. Light gray: solver-computed outputs $\psi_t$. Crimson: approximated segments. **Left—Reuse only:** numerically stable but limited in expressivity; fine details erode (bottom). **Middle—Predictor only:** chaining second-order extrapolations overshoots without re-anchoring, producing artifacts (bottom). **Right—Reuse observed information (ZEUS):** alternating reuse of $\{\psi_t,\ \psi_t + \Delta^{(1)}\psi_t\}$ prevents overshoot and preserves detail, yielding the best perceptual quality.

typically inserting a single fresh denoiser call between several approximated steps. This creates a persistent scarcity of fresh computation. Under this constraint, we require an approximation that (i) *uses all signals that are already **fully-computed** along the executed trajectory* and (ii) *stays principled and minimal.*

At a given fresh step $t$, we assume the neighbors $t+1$ and $t-1$ are reduced and approximated, along with the state's denoiser output $\psi_t$ fully computed. This leads to the "reuse only" scheme in previous works Ma et al. (2024), where the next reduced model output is approximated by the fresh model output $\hat{\psi}_{t-1} \leftarrow \psi_t$. This scheme yields non-distorting results, yet suffers from low similarity and degraded details compared to the original sample, as shown in Fig.4b

Crucially, we have already advanced the solver from $\hat{x}_{t+1}$ to $\hat{x}_t$ using the previously available signal $\hat{\psi}_{t+1}$, and then evaluated the denoiser to obtain the fresh $\psi_t$ at $(\hat{x}_t, t)$. Hence, the backward first-order difference $\Delta^{(1)}\psi_t := \psi_t - \hat{\psi}_{t+1}$ is an *observed, path-wise* quantity: it captures the model's realized change in output between the received signal at $t+1$ and the fresh computation at $t$. Under scarcity, the information set at $t$ is $\{\psi_t,\ \Delta^{(1)}\psi_t\}$. This naturally introduces a finite difference predictor in second order $\hat{\psi}_{t-1} \leftarrow 2\psi_t - \hat{\psi}_{t+1}$, which can be regarded as an inductive field of the available information set at $t$, as visualized by Fig. 2. This predictor underlies several recent training-free accelerations Liu et al. (2025); Yu et al. (2025) and we adopt it as the *minimal, information-consistent* choice under scarce fresh computation.

> **Theorem 4.1** (unique tuple under $2\times$ acceleration). *If the scheme achieves a speed-up factor of at least $2\times$, then by the pigeonhole principle each local window $\{t-1, t, t+1\}$ contains at most one full network evaluation. Hence the only genuinely available computational unit is the 2-tuple*
>
> $$(\psi_t,\ \Delta^{(1)}\psi_t),$$
>
> *consisting of one evaluated state $\psi_t$ and its deterministic difference $\Delta^{(1)}\psi_t$.*

A seemingly natural upgrade is a third-order predictor. However, it would yield suboptimal results, as shown in Fig. 3, as any higher-order scheme necessarily substitutes additional *approximated* points, inflating the upstream term and magnifying accumulated error. In the next paragraph, we rigorously demonstrate the surprising effectiveness of a second-order predictor.

**The surprising effectiveness of a second-order predictor** Under scarce fresh computation, this pair is the entire available signal at a fresh step $t$, and everything else (e.g., $\hat{\psi}_{t+1}$) is an algebraic

reuse with no new network signal. The "reuse-only" rule $\hat{\psi}_{t-1} \leftarrow \psi_t$ honors this constraint but throws away the trend $\Delta^{(1)}\psi_t$, yielding a zeroth-order hold that cannot cancel the (typically large) affine drift of $\psi_\theta(\cdot, t)$ along the executed trajectory. In contrast, the *second-order* backward predictor

$$\hat{\psi}_{t-1} \; = \; 2\,\psi_t - \hat{\psi}_{t+1} \; = \; \psi_t + \Delta^{(1)}\psi_t \tag{2}$$

uses *exactly* the freshly evaluated signal $\psi_t$ and the already realized first difference $\Delta^{(1)}\psi_t$, adding no extra model calls and no additional state besides what the solver has already materialized. This simple rule is principled in four complementary senses.

*(i) Affine invariance and BLUE optimality.* Writing the denoiser outputs locally as $\psi_u = \phi(u) + \eta_u$ with a smooth trend $\phi$ and zero-mean perturbation $\eta$ (Theorem A.4), the weights $(2, -1)$ in equation 2 are the *unique* linear coefficients that are unbiased for all affine $\phi$ and, under homoscedastic uncorrelated perturbations, minimize variance among all such estimators—i.e., equation 2 is the BLUE (Theorem A.5).

*(ii) Second-order accuracy and minimax sufficiency.* A Taylor expansion at $t$ shows the local truncation error $\mathbb{E}[\psi_{t-1} - (2\psi_t - \hat{\psi}_{t+1})] = \Delta^2\,\phi''(s_t) + o(\Delta^2)$ so the bias is $O(\Delta^2)$ (Theorem A.5). Moreover, without additional fresh evaluations the $\Delta^2$ rate is *information-theoretically optimal*: any estimator using only $\{\psi_t, \hat{\psi}_{t+1}\}$ must incur $\Omega(\Delta^2)$ worst-case bias over $C^2$ trends (Theorem A.10), which equation 2 attains tightly (Theorem A.11). By contrast, one-point reuse is provably limited to $\Theta(\Delta)$ error, explaining its detail loss and low similarity.

*(iii) Curvature awareness without extra cost.* The predictor equation 2 implicitly measures the second difference $\Delta^2\phi''(s_t)$, so it is responsive to local bending of the denoiser's response while remaining insensitive to global shifts or linear ramps. This "curvature gating" is precisely what preserves fine details when skipping steps.

*(iv) Stability and parameterization invariance.* Because all common parameterizations ($\epsilon$, $\mathbf{x}_0$, $v$, score, flow) are related by fixed affine readouts in $s$, equation 2 commutes with these transformations, making the rule architecture- and target-agnostic. At the same time, higher-order extrapolants necessarily ingest more *approximated* points and inflate variance with rapidly growing weights, leading to noise amplification and reduced stability under uniform time grids, while offering *no* minimax bias improvement beyond $O(\Delta^2)$ in our $C^2$ regime (Section A.3.4).

### 4.2 How can we approximate multiple consecutive steps?

In this section, we aim to reduce computations in as many consecutive states as possible in a numerically stable manner. Consider $k \geq 2$ consecutive reduced steps starting with $\psi_{t-1}$ in the denoising process. A second-order predictor tends to overshoot (Fig. 4b), as chaining extrapolants compounds approximation error without re-anchoring. By comparison, a reuse-only baseline tends to prevent overshooting but underutilizes available information, leading to detail loss (Fig. 4b). This exposes a trade-off between the approximation precision from the second-order predictor and the approximation stability from the reuse-only baseline. A question naturally arises: ***Can we improve both precision and stability at the same time?***

In this work, we offer a simple but effective solution: **we reuse the observed information set** $\{\psi_t, \Delta^{(1)}\psi_t\}$. As illustrated in Figure 4a, we duplicate the two-element tuple $\{\psi_t, \hat{\psi}_{t-1} \leftarrow \psi_t + \Delta^{(1)}\psi_t\}$, forming a "zig-zag" pattern. When reducing computation for multiple consecutive steps, reusing the tuple preserves approximation stability, and leveraging all fully-computed information (i.e., the tuple) enhances approximation precision. We then provide a rigorous analysis of the induced error of the three strategies above.

In particular, their bias–variance behavior can be precisely characterized as follows. Consistent with Theorems A.19 and A.20, two-point 2-nd order prediction achieves second-order bias, $\|\mathrm{Bias}(\hat{\psi}_{t-j})\| = O(j^2\Delta^2)$, with the exact expansion given in Theorem A.19, but its variance grows quadratically with the jump length, $\mathrm{Var}(\hat{\psi}_{t-j}) = \big((j+1)^2 + j^2\big)\sigma^2 I_d$.

In contrast, reuse observed and reuse only maintain a $j$-independent variance (equal to $\sigma^2 I_d$ for even-$j$ reuse observed and reuse only, and $5\sigma^2 I_d$ for odd-$j$ reuse observed), but remain only first-order accurate with $\|\mathrm{Bias}\| = O(j\Delta)$.

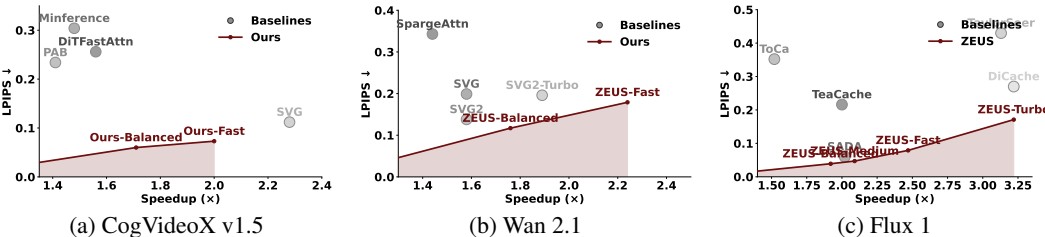

(a) CogVideoX v1.5      (b) Wan 2.1      (c) Flux 1

Figure 5: **Scaling: Speedup vs. LPIPS ($\downarrow$).** Gray circles denote training-free baselines; the maroon polyline marks our ZEUS variants (Balanced→Medium→Fast→Turbo). Rightward is faster, downward is better. Across CogVideoX-v1.5, Wan-2.1, and Flux (Euler/flow), *ZEUS exhibits near-linear scaling*—consistently attaining high speedups with lower LPIPS.

Our reuse observed scheme, built on the pair $\{\psi_t, \Delta^{(1)}\psi_t\}$, inherits the best property at the *first* reduced step: when $j{=}1$, it coincides with the $(2, -1)$ BLUE, attaining second-order bias $O(\Delta^2)$ while keeping constant variance $5\sigma^2 I_d$. For longer jumps, duplicating the tuple preserves the constant-variance behavior of reuse only, yet remains bias-dominated $O(j\Delta)$ rather than variance-dominated. Consequently, in low-noise, short-jump regimes the one-step estimate achieves strictly smaller MSE; as jump length increases or noise grows, 2-nd order prediction suffers variance explosion, whereas reuse observed remains numerically stable—2-nd order prediction tends to overshoot, while reuse observed stays stable at the cost of precision. A full derivation of these results is provided in Section A.5.

## 5 EXPERIMENT

### 5.1 EXPERIMENTAL SETTINGS

**Models Configuration.** To assess generalization across modalities, architectures, and prediction objectives, we evaluate ZEUS on **Text-to-image** and **Text-to-video** models in a comprehensive and diverse setting. *Image:* Stable Diffusion 2 (U-Net, $v$-prediction) Rombach et al. (2022), SDXL (modified U-Net, $\epsilon$-prediction) Podell et al., and Flux.1-dev (MMDiT, $u$-flow matching) Black-Forest-Labs (2024). *Video:* Wan2.1-T2V-14B (DiT, $u$-flow matching) Wan et al. (2025) and CogVideoX-v1.5-T2V (DiT, $\epsilon$-prediction) Yang et al. (2025b). We use two standard ODE solvers—*Euler* (first order) Karras et al. (2022) and *DPM-Solver++* (second order) Lu et al. (2022a;b)—with 50 sampling steps in all main experiments. All pipelines are implemented in the HuggingFace `diffusers` framework for reproducibility, and all runs use a single NVIDIA A100 (80 GB) GPU.

**Evaluation Metrics** Our objective is to *preserve baseline fidelity while reducing latency*. For *text-to-image*, we follow MS COCO–2017 captions as prompts Lin et al. (2014) under identical seeds and guidance. Efficiency is reported as the end-to-end wall-clock speedup relative to the baseline pipeline. Generation quality is evaluated using the Peak Signal-to-Noise Ratio (PSNR), Learned Perceptual Image Patch Similarity (LPIPS) Zhang et al. (2018), and Fréchet Inception Distance (FID) Heusel et al. (2017) between original generated and accelerated samples. For *Text-to-video* experiments, we follow the exact setup in the SparseVideo Gen series Xi et al. (2025); Yang et al. (2025a). We adopt the VBench/Penguin prompts provided by the VBench team. Generation quality is evaluated using LPIPS and FID, alongside Structural Similarity Index Measure (SSIM).

**Baselines** We compare ZEUS to a comprehensive list of recent training-free acceleration strategies, to the best of our knowledge. For *Text-to-image*, We compare against DeepCache Ma et al. (2024), AdaptiveDiffusion Ye et al. (2024), SADA Jiang et al. (2025) on stable diffusion models with U-Net backbones. We compare against ToCa Zou et al. (2024), TaylorSeer , TeaCache Liu et al. (2024), DiCache Bu et al. (2025), and SADA on FLUX.1-dev with MMDiT backbone. For *Text-to-video*, we adopt the evaluation suite from the Sparse VideoGen series Xi et al. (2025); Yang et al. (2025a). We compare against DiTFastAttn Yuan et al. (2024), Minference Jiang et al. (2024), PAB Zhao et al. (2024), and SpargeAttn Zhang et al. (2025b) along with the SVG series and TeaCache.

Table 2: **Image quality vs. speed on SD-2, SDXL, and FLUX.** Higher is better for PSNR/Speedup; lower is better for LPIPS/FID. Per *model+scheduler* block, best is **bold**, second best is underlined. ZEUS variants are highlighted only from the *Method* column onward.

| Model | Scheduler | Method | PSNR ↑ | LPIPS ↓ | FID ↓ | Speedup ↑ |
|---|---|---|---|---|---|---|
| **SD-2** | DPM++ | DeepCache | 17.7 | 0.271 | 7.83 | 1.43× |
| | | AdaptiveDiffusion | 24.3 | 0.100 | 4.35 | 1.45× |
| | | SADA | 26.34 | 0.094 | 4.02 | 1.80× |
| | | **ZEUS-Quality** | **28.37** | **0.0641** | **2.95** | 1.56× |
| | | **ZEUS-Medium** | 25.72 | 0.1039 | 4.46 | **1.85×** |
| | Euler | DeepCache | 18.9 | 0.239 | 7.40 | 1.45× |
| | | AdaptiveDiffusion | 21.9 | 0.173 | 7.58 | **1.89×** |
| | | SADA | 26.25 | 0.100 | 4.26 | 1.81× |
| | | **ZEUS-Quality** | **27.35** | **0.078** | **3.57** | 1.57× |
| | | **ZEUS-Medium** | 25.37 | 0.118 | 5.06 | 1.86× |
| **SDXL** | DPM++ | DeepCache | 21.3 | 0.255 | 8.48 | 1.74× |
| | | AdaptiveDiffusion | 26.1 | 0.125 | 4.59 | 1.65× |
| | | SADA | 29.36 | 0.084 | 3.51 | 1.86× |
| | | **ZEUS-Quality** | **31.38** | **0.058** | **2.57** | 1.57× |
| | | **ZEUS-Medium** | 29.17 | 0.084 | 3.59 | 1.87× |
| | | **ZEUS-Fast** | 26.38 | 0.129 | 5.39 | **1.93×** |
| | Euler | DeepCache | 22.00 | 0.223 | 7.36 | **2.16×** |
| | | AdaptiveDiffusion | 24.33 | 0.168 | 6.11 | 2.01× |
| | | SADA | 28.97 | 0.093 | 3.76 | 1.85× |
| | | **ZEUS-Quality** | **30.25** | **0.071** | **3.02** | 1.57× |
| | | **ZEUS-Medium** | 28.66 | 0.095 | 3.87 | 1.85× |
| | | **ZEUS-Fast** | 25.15 | 0.153 | 6.47 | 1.93× |
| **Flux** | Euler (Flow) | TeaCache | 19.14 | 0.216 | 4.89 | 2.00× |
| | | SADA | 29.44 | 0.060 | 1.95 | 2.02× |
| | | ToCa | 17.70 | 0.352 | 8.84 | 1.52× |
| | | TaylorSeer | 15.36 | 0.430 | 10.08 | 3.13× |
| | | DiCache | 22.39 | 0.270 | / | 3.22× |
| | | **ZEUS-Balanced** | **31.08** | **0.039** | **1.29** | 1.92× |
| | | **ZEUS-Medium** | 30.19 | 0.047 | 1.53 | 2.09× |
| | | **ZEUS-Fast** | 26.77 | 0.079 | 2.49 | 2.47× |
| | | **ZEUS-Turbo** | 21.80 | 0.171 | 4.52 | 3.22× |

## 5.2 MAIN RESULTS

**Image Results.** We examine **ZEUS** on modern diffusion stacks (Flux.v1, SDXL, SD-2/1.5). With only a minor, training-free change to the sampler, ZEUS consistently moves the speed–quality frontier outward. On classic ODE schedules (e.g., DPM++), ZEUS variants reach the fastest end-to-end runtimes while simultaneously improving perceptual fidelity—for example on **SDXL/DPM++**, *ZEUS-Fast* attains the top speed (about 1.93×) while *ZEUS-Quality* improves LPIPS/FID over the strongest baseline (e.g., 0.058/2.57 vs. 0.084/3.51). On flow-matching with **Flux**, the gap is larger, with *ZEUS-Turbo* delivering markedly higher throughput (about 3.22× vs. 2.02×) and *ZEUS-Balanced* achieving stronger LPIPS/FID (e.g., 0.039/1.29 vs. 0.060/1.95). Even in the few settings where another method edges out ZEUS on raw speed, the *ZEUS-Quality* variant still attains the best perceptual scores, and the *ZEUS-Fast*/*ZEUS-Medium* variants trace a smooth Pareto curve between quality and speed. Overall, a small, model-agnostic, training-free tweak to step selection and cache-aware reconstruction yields state-of-the-art or near-SOTA speed while *consistently* preserving perceptual realism.

**Video results.** Across both **Wan 2.1** and **CogVideoX-v1.5**, **ZEUS** establishes a new quality–efficiency frontier *without* any video-specific kernels or model changes (Tab. 3). In the *Balanced* setting on Wan 2.1, ZEUS delivers higher perceptual fidelity (LPIPS↓/PSNR↑: 0.117/29) while still running faster than the strongest prior (1.76× vs. 1.58×), and it remains near the top on SSIM. Push-

Table 3: **Quality & efficiency on video generation.** Higher is better for PSNR/SSIM/Speedup; lower is better for LPIPS. Per-*model* best is **bold**, second-best is underlined.

| Model | Method | PSNR ↑ | SSIM ↑ | LPIPS ↓ | Speedup ↑ |
|-------|--------|--------|--------|---------|-----------|
| **Wan 2.1** | | | | | |
| | SpargeAttn | 20.519 | 0.623 | 0.343 | 1.44× |
| | SVG | 22.989 | 0.785 | 0.199 | 1.58× |
| | SVG2 | 25.808 | **0.854** | 0.138 | 1.58× |
| | SVG2-Turbo | 23.682 | 0.789 | 0.196 | 1.89× |
| | Ours-Balanced | **29.000** | 0.846 | **0.117** | 1.76× |
| | Ours-Fast | 26.594 | 0.7919 | 0.179 | **2.24×** |
| **CogVideoX-v1.5** | | | | | |
| | DiTFastAttn | 23.202 | 0.741 | 0.256 | 1.56× |
| | Minference | 22.451 | 0.691 | 0.304 | 1.48× |
| | PAB | 22.486 | 0.740 | 0.234 | 1.41× |
| | SVG | 29.989 | **0.910** | 0.112 | **2.28×** |
| | Ours-Balanced | **32.495** | 0.893 | **0.060** | 1.71× |
| | Ours-Fast | 30.970 | 0.876 | 0.073 | 2.00× |

ing to the *Fast* setting, ZEUS sustains top-tier throughput ( 2.24×) with competitive LPIPS/SSIM, outperforming sparsity baselines at comparable cost. The same trend holds on CogVideoX-v1.5: *Balanced* achieves the best perceptual scores at healthy speed ( 1.7×), while *Fast* reaches 2× and stays close to the quality frontier. In short, a single, model-agnostic skipping policy consistently lets us *raise fidelity at higher speed* or *raise speed at better fidelity* across both T2V generators, dominating the quality–latency trade-off. Although minimal, ZEUS achieves significantly better similarity consistently across experiment settings, as illustrated in Fig. 5.

## 5.3 ABLATION STUDIES

Table 4: Ablation study on few-step sampling across schedulers. Results on MS-COCO 2017.

| Flux | | | | |
|------|------|--------|-------|---------|
| Scheduler | Steps | LPIPS ↓ | FID ↓ | Speedup |
| Euler | 50 | 0.047 | 1.53 | 2.09× |
| | 25 | 0.040 | 2.65 | 1.58× |
| | 15 | 0.036 | 2.47 | 1.22× |
| **SDXL** | | | | |
| Euler | 50 | 0.095 | 3.87 | 1.85× |
| | 25 | 0.064 | 4.98 | 1.46× |
| | 15 | 0.065 | 4.72 | 1.20× |
| DPM++ | 50 | 0.084 | 3.59 | 1.87× |
| | 25 | 0.066 | 5.28 | 1.45× |
| | 15 | 0.066 | 4.87 | 1.20× |

We conduct ablation studies on few-step sampling. Table 4 reports few-step sampling on MS-COCO 2017 across schedulers and backbones with steps $\{50, 25, 15\}$. For **Flux** (Euler), ZEUS lowers LPIPS from 0.047 to 0.036 as steps drop from 50 to 15, with FID moving from 1.53 to 2.47, while achieving 2.09×, 1.58×, and 1.22× speedups, respectively. For **SDXL**, ZEUS shows the same pattern: with *Euler*, LPIPS improves from 0.095 to 0.065 as steps decrease, with FID ranging $3.87 \to 4.72$, and speedups 1.85×, 1.46×, 1.20×; with *DPM++*, LPIPS improves from 0.084 to 0.066, FID ranges $3.59 \to 4.87$, and speedups are 1.87×, 1.45×, 1.20×. Overall, ZEUS consistently preserves or improves perceptual similarity (LPIPS) under aggressive step reduction, with modest FID trade-offs, and delivers stable acceleration in the few-step regime (about 1.5× at 25 steps and 1.2× at 15 steps) across backbones and schedulers.

## 6 CONCLUSION

In this paper, we introduce ZEUS: a minimal, training-free, method-agnostic plug-in that consistently shifts the speed–fidelity Pareto frontier across five backbones and two solvers—achieving up to 3.2× end-to-end speedup on Flux while improving LPIPS/FID/PSNR. ZEUS addresses the scarcity of fresh computation in an ambitious acceleration scenario. Reusing and leveraging the observed information set from this constraint, yielding either better quality at the same cost or higher speed at comparable quality. ZEUS conveys a counterintuitive but compelling message: ***accelerating diffusion is as easy as a second-order predictor.***

## LLMs Usage Statement

We clarify that LLMs were used solely as auxiliary aids, restricted to refining the manuscript's exposition for clarity and conciseness.

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

CONTENTS

# Appendix

## A MATHEMATICAL FOUNDATIONS

### A.1 NOTATION

In this section, we firstly formalize all the specific notation used in the paper.

**Definition A.1.** Let $(\Omega, \mathcal{F}, \mathrm{Pr})$ be the probability space that generates the random pair $(x_0, \boldsymbol{\epsilon})$ and hence the noisy latent $x_t = \alpha_t x_0 + \sigma_t \boldsymbol{\epsilon}$ at any $t \in \{1, \ldots, T\}$. For a fixed, deterministic network $\psi_\theta(\cdot, \cdot) : \mathbb{R}^d \times [0, 1] \to \mathbb{R}^d$, which is the network training result for $\psi_t$ like $\boldsymbol{\epsilon}$ or $v$ in the forward process, define the random output $\psi_{\theta, t} := \psi_\theta(x_t, t) \in L^2(\Omega)$ and define the output in inference process $\hat{\psi}_t := \psi_\theta(\hat{x}_t, t)$.

**General forward process.** Let $(\Omega, \mathcal{F}, \mathrm{Pr})$ be a probability space. A data sample is represented by a random vector $\mathbf{x}_0 : \Omega \to \mathbb{R}^d$ with distribution $p_{\mathrm{data}}$. Let $\boldsymbol{\epsilon} : \Omega \to \mathbb{R}^d$ be an independent standard Gaussian noise, i.e. $\boldsymbol{\epsilon} \sim \mathcal{N}(\mathbf{0}, \mathbf{I}_d)$. We distinguish continuous time $s \in [0, 1]$ and a uniform discrete grid $s_t := t/T$ for $t \in \{0, 1, \ldots, T\}$ with step size $\Delta := 1/T$. For each $s \in [0, 1]$, define deterministic schedule functions $\alpha_s \in (0, 1]$, $\sigma_s \in (0, 1]$.

The forward latent at time $s$ is defined as

$$\mathbf{x}_s := \alpha_s \mathbf{x}_0 + \sigma_s \boldsymbol{\epsilon}, \qquad \mathbf{x}_s \in L^2(\Omega; \mathbb{R}^d). \tag{A.1}$$

Equivalently, conditioned on $\mathbf{x}_0$, the marginal distribution is Gaussian:

$$q(\mathbf{x}_s \mid \mathbf{x}_0) = \mathcal{N}(\alpha_s \mathbf{x}_0, \sigma_s^2 \mathbf{I}_d).$$

Thus the diffusion forward process is the family $\{\mathbf{x}_s : s \in [0, 1]\}$, with discrete samples $\{\mathbf{x}_t : t = 0, \ldots, T\}$ obtained by evaluating at time grid points.

**Reverse process.** The forward process $\{\mathbf{x}_s : s \in [0, 1]\}$ defined in equation A.1 is Markovian. In particular, for any $0 \leq s' < s \leq 1$, the conditional distribution of $\mathbf{x}_{s'}$ given $(\mathbf{x}_s, \mathbf{x}_0)$ is Gaussian:

$$q(\mathbf{x}_{s'} \mid \mathbf{x}_s, \mathbf{x}_0) = \mathcal{N}(\mu_{s', s}(\mathbf{x}_s, \mathbf{x}_0), \ \Sigma_{s', s}), \tag{A.2}$$

with mean

$$\mu_{s', s}(\mathbf{x}_s, \mathbf{x}_0) = \frac{\alpha_{s'}}{\alpha_s} \mathbf{x}_s + \left(\alpha_0 - \frac{\alpha_{s'}}{\alpha_s} \alpha_s\right) \mathbf{x}_0 = \frac{\alpha_{s'}}{\alpha_s} \mathbf{x}_s + \left(1 - \frac{\alpha_{s'}}{\alpha_s}\right) \mathbf{x}_0,$$

and covariance

$$\Sigma_{s', s} = \left(\sigma_{s'}^2 - \frac{\alpha_{s'}^2}{\alpha_s^2} \sigma_s^2\right) \mathbf{I}_d.$$

Equivalently, marginalizing out $\mathbf{x}_0$, the reverse-time transition kernel can be expressed as

$$q(\mathbf{x}_{s'} \mid \mathbf{x}_s) = \int q(\mathbf{x}_{s'} \mid \mathbf{x}_s, \mathbf{x}_0) \, p(\mathbf{x}_0 \mid \mathbf{x}_s) \, d\mathbf{x}_0, \tag{A.3}$$

which is generally intractable. The role of the neural network is precisely to approximate the conditional dependence on $\mathbf{x}_0$ by predicting either the noise $\boldsymbol{\epsilon}$, the clean sample $\mathbf{x}_0$, or equivalent parameterizations.

**Training objective.** The network is trained by conditional regression: for a chosen ground-truth target $\psi_0(\mathbf{x}_0, \boldsymbol{\epsilon}, s)$, we minimize

$$\min_\theta \ \mathbb{E}_{\mathbf{x}_0 \sim p_{\mathrm{data}}, \, \boldsymbol{\epsilon} \sim \mathcal{N}(0, I), \, s \sim \mathcal{U}[0, 1]} \left[\ell\big(\psi_\theta(\mathbf{x}_s, s), \ \psi_0(\mathbf{x}_0, \boldsymbol{\epsilon}, s)\big)\right],$$

where $\ell$ is typically the squared error.

**Unified network parameterization.** To express all variants within a single notation, we define

$$\psi_\theta : \mathbb{R}^d \times [0,1] \to \mathbb{R}^d, \qquad \psi_\theta(\mathbf{x}_s, s) \text{ trained to predict } \psi_0(\mathbf{x}_0, \boldsymbol{\epsilon}, s).$$

Here $\psi_0$ is chosen from a finite family of equivalent targets, e.g.

$$\psi_0 \in \left\{ \boldsymbol{\epsilon}, \ \mathbf{x}_0, \ \alpha_s \boldsymbol{\epsilon} - \sigma_s \mathbf{x}_0, \ \nabla_{\mathbf{x}_s} \log q_s(\mathbf{x}_s), \ \boldsymbol{\epsilon} - \mathbf{x}_0 \right\},$$

corresponding to $\epsilon$-prediction, $\mathbf{x_0}$-prediction, $v$-prediction, score-prediction, and flow-matching. Thus, different implementations correspond to linearly related instances of the same abstract map $\psi_\theta$, which enables uniform analysis of loss functions and inference rules.

**Continuous-time probability flow ODE.** The forward process $\{\mathbf{x}_s : s \in [0,1]\}$ in equation A.1 admits a continuous-time formulation as a linear Itô SDE:

$$d\mathbf{x}_s \ = \ f(s)\, \mathbf{x}_s\, ds + g(s)\, d\mathbf{w}_s, \tag{A.4}$$

where $\mathbf{w}_s$ is a standard Wiener process in $\mathbb{R}^d$, and the drift/diffusion coefficients $(f(s), g(s))$ are determined by the schedules $(\alpha_s, \sigma_s)$. Concretely, the marginal law of $\mathbf{x}_s$ given $\mathbf{x}_0$ is Gaussian with mean $\alpha_s \mathbf{x}_0$ and variance $\sigma_s^2 \mathbf{I}_d$.

Following (Song et al., b; Song & Ermon, 2019; Chen et al., 2022), the reverse-time SDE that generates the same marginal distributions runs backward from $s = 1$ to $s = 0$:

$$d\mathbf{x}_s \ = \ \left[ f(s)\, \mathbf{x}_s - g(s)^2\, \nabla_{\mathbf{x}_s} \log q_s(\mathbf{x}_s) \right] ds + g(s)\, d\bar{\mathbf{w}}_s, \tag{A.5}$$

where $q_s$ denotes the density of $\mathbf{x}_s$, and $\bar{\mathbf{w}}_s$ is a standard Wiener process running backward in time.

The *probability flow ODE* (PF-ODE) is the deterministic counterpart of equation A.5, obtained by removing the stochastic term:

$$d\mathbf{x}_s \ = \ \left[ f(s)\, \mathbf{x}_s - \tfrac{1}{2} g(s)^2\, \nabla_{\mathbf{x}_s} \log q_s(\mathbf{x}_s) \right] ds. \tag{A.6}$$

This ODE preserves the exact marginal distributions $\{q_s\}_{s \in [0,1]}$ of the forward process, and therefore provides a deterministic generative sampling procedure.

## A.2 SOME THEOREM

**Consistency of the training objective.** Recall that the network is trained by conditional regression:

$$\min_{\theta} \ \mathbb{E}_{\mathbf{x}_0 \sim p_{\text{data}}, \ \boldsymbol{\epsilon} \sim \mathcal{N}(0, I), \ s \sim \mathcal{U}[0,1]} \big[ \ell\big( \psi_{\theta}(\mathbf{x}_s, s), \ \psi_0(\mathbf{x}_0, \boldsymbol{\epsilon}, s) \big) \big],$$

where $\mathbf{x}_s = \alpha_s \mathbf{x}_0 + \sigma_s \boldsymbol{\epsilon}$ and $\ell$ is the squared error loss.

> **Theorem A.2** (Optimal predictor under $L^2$ training). *Let $\ell(a, b) = \|a - b\|_2^2$. Define the population risk*
>
> $$\mathcal{L}(\theta) := \mathbb{E}_{\mathbf{x}_0, \boldsymbol{\epsilon}, s} \big[ \| \psi_{\theta}(\mathbf{x}_s, s) - \psi_0(\mathbf{x}_0, \boldsymbol{\epsilon}, s) \|_2^2 \big].$$
>
> *Then any minimizer $\psi^*$ of $\mathcal{L}$ satisfies, for all $(\mathbf{x}_s, s)$,*
>
> $$\psi^*(\mathbf{x}_s, s) = \mathbb{E}\big[ \psi_0(\mathbf{x}_0, \boldsymbol{\epsilon}, s) \mid \mathbf{x}_s, s \big]. \tag{A.7}$$
>
> *In other words, in the $L^2$ sense, training recovers the conditional expectation of the regression target $\psi_0$ given the noisy input $(\mathbf{x}_s, s)$. Let the hypothesis class be all measurable maps with finite second moment; equivalently, consider the Bayes risk minimization."*

*Proof.* For fixed $(\mathbf{x}_s, s)$, define the conditional distribution

$$p(\mathbf{x}_0, \boldsymbol{\epsilon} \mid \mathbf{x}_s, s).$$

The contribution of $(\mathbf{x}_s, s)$ to the expected loss is

$$\mathbb{E}_{\mathbf{x}_0, \boldsymbol{\epsilon} \mid \mathbf{x}_s, s} \Big[ \| \psi_{\theta}(\mathbf{x}_s, s) - \psi_0(\mathbf{x}_0, \boldsymbol{\epsilon}, s) \|_2^2 \Big].$$

This is a convex quadratic in $\psi_{\theta}(\mathbf{x}_s, s)$, uniquely minimized at

$$\psi^*(\mathbf{x}_s, s) = \mathbb{E}\big[ \psi_0(\mathbf{x}_0, \boldsymbol{\epsilon}, s) \mid \mathbf{x}_s, s \big].$$

Therefore, the global risk minimizer $\psi^*$ coincides with the conditional expectation equation A.7. $\square$

**Discussion.** The theorem shows that, under exact optimization of the $L^2$ objective, the learned network $\psi_{\theta}$ does not in general recover the ground-truth target $\psi_0(\mathbf{x}_0, \boldsymbol{\epsilon}, s)$ pointwise. Instead, it recovers its conditional expectation given the accessible input $(\mathbf{x}_s, s)$. Thus, the choice of $\psi_0$ (noise, clean data, or equivalent parameterizations) directly determines which conditional expectation is realized by the trained model.

**Unified network parameterization.** We now establish that all common training targets in diffusion-type models are instances of the same conditional regression principle.

> **Theorem A.3** (Equivalence of parameterizations). *Let the forward process follow Definition A.1, i.e. $\mathbf{x}_s = \alpha_s \mathbf{x}_0 + \sigma_s \boldsymbol{\epsilon}$ with $\boldsymbol{\epsilon} \sim \mathcal{N}(0, I)$. Fix a target functional $\psi_0(\mathbf{x}_0, \boldsymbol{\epsilon}, s)$ from a finite set of admissible forms. Suppose the network is trained with the $L^2$ objective*
>
> $$\mathcal{L}(\theta) = \mathbb{E}_{\mathbf{x}_0, \boldsymbol{\epsilon}, s} \big[ \| \psi_{\theta}(\mathbf{x}_s, s) - \psi_0(\mathbf{x}_0, \boldsymbol{\epsilon}, s) \|^2 \big].$$
>
> *Then, in the limit of exact optimization, the optimal predictor satisfies*
>
> $$\psi^*(\mathbf{x}_s, s) = \mathbb{E}\big[ \psi_0(\mathbf{x}_0, \boldsymbol{\epsilon}, s) \mid \mathbf{x}_s, s \big].$$
>
> *Moreover, for each admissible $\psi_0$, there exists deterministic coefficients $(a_s, b_s)$ such that the clean data is exactly recovered by*
>
> $$\hat{\mathbf{x}}_0^{(s)} = a_s \mathbf{x}_s + b_s \psi^*(\mathbf{x}_s, s). \tag{A.8}$$
>
> *Thus, all parameterizations are equivalent in expressive power: they differ only in the choice of regression target $\psi_0$ and in the reconstruction formula equation A.12.*

*Proof.* From Theorem A.2, the $L^2$ minimizer satisfies

$$\psi^*(\mathbf{x}_s, s) \ = \ \mathbb{E}[\psi_0(\mathbf{x}_0, \boldsymbol{\epsilon}, s) \mid \mathbf{x}_s, s].$$

By construction, the forward process is linear-Gaussian:

$$\mathbf{x}_s = \alpha_s \mathbf{x}_0 + \sigma_s \boldsymbol{\epsilon}, \qquad \boldsymbol{\epsilon} \sim \mathcal{N}(\mathbf{0}, \mathbf{I}), \; \boldsymbol{\epsilon} \perp\!\!\!\perp \mathbf{x}_0. \tag{A.9}$$

Each admissible training target is an affine (here linear) functional of $(\mathbf{x}_0, \boldsymbol{\epsilon})$. We unify them by writing

$$\psi_0(\mathbf{x}_0, \boldsymbol{\epsilon}, s) = u_s \boldsymbol{\epsilon} + v_s \mathbf{x}_0, \tag{A.10}$$

where $(u_s, v_s)$ are scalar (or diagonal, coordinate-wise) coefficients depending only on $s$. Combining equation A.9–equation A.10,

$$\begin{bmatrix} \mathbf{x}_s \\ \psi_0 \end{bmatrix} = \begin{bmatrix} \alpha_s & \sigma_s \\ v_s & u_s \end{bmatrix} \begin{bmatrix} \mathbf{x}_0 \\ \boldsymbol{\epsilon} \end{bmatrix} := \mathbf{M}_s \begin{bmatrix} \mathbf{x}_0 \\ \boldsymbol{\epsilon} \end{bmatrix}.$$

Assume $\det(\mathbf{M}_s) \neq 0$, i.e. $\Delta_s := \alpha_s u_s - \sigma_s v_s \neq 0$. Then $\mathbf{M}_s$ is invertible and we have the *algebraic identity*

$$\mathbf{x}_0 = a_s \mathbf{x}_s + b_s \psi_0, \qquad a_s = \frac{u_s}{\Delta_s}, \quad b_s = -\frac{\sigma_s}{\Delta_s}. \tag{A.11}$$

Taking conditional expectation of equation A.11 given $(\mathbf{x}_s, s)$ and using $\psi^* = \mathbb{E}[\psi_0 \mid \mathbf{x}_s, s]$, we obtain

$$\mathbb{E}[\mathbf{x}_0 \mid \mathbf{x}_s, s] = a_s \mathbf{x}_s + b_s \psi^*(\mathbf{x}_s, s). \tag{A.12}$$

Thus the reconstruction rule $\hat{\mathbf{x}}_0^{(s)} := a_s \mathbf{x}_s + b_s \psi^*(\mathbf{x}_s, s)$ exactly equals the posterior mean $\mathbb{E}[\mathbf{x}_0 \mid \mathbf{x}_s, s]$; when $\psi^* \equiv \psi_0$ (ideal limit), equation A.11 is a pointwise identity. In particular, $\hat{\mathbf{x}}_0^{(s)}$ is an *unbiased estimator* of the clean sample $\mathbf{x}_0$. Therefore, any admissible $\psi_0$ induces a unique pair $(a_s, b_s)$ and an equivalent reconstruction of $\mathbf{x}_0$.

**Instantiations.** We now instantiate equation A.11 for the common parameterizations by plugging the corresponding $(u_s, v_s)$:

1. **$\boldsymbol{\epsilon}$-prediction:** $\psi_0 = \boldsymbol{\epsilon}$, i.e. $(u_s, v_s) = (1, 0)$. Then $\Delta_s = \alpha_s$ and

$$a_s = \frac{1}{\alpha_s}, \quad b_s = -\frac{\sigma_s}{\alpha_s}, \qquad \Rightarrow \quad \hat{\mathbf{x}}_0^{(s)} = \frac{1}{\alpha_s} \mathbf{x}_s - \frac{\sigma_s}{\alpha_s} \psi^*(\mathbf{x}_s, s).$$

2. **$\mathbf{x}_0$-prediction:** $\psi_0 = \mathbf{x}_0$, i.e. $(u_s, v_s) = (0, 1)$. Then $\Delta_s = -\sigma_s$ and

$$a_s = 0, \quad b_s = 1, \qquad \Rightarrow \quad \hat{\mathbf{x}}_0^{(s)} = \psi^*(\mathbf{x}_s, s).$$

3. **$v$-prediction:** $\psi_0 = \alpha_s \boldsymbol{\epsilon} - \sigma_s \mathbf{x}_0$, i.e. $(u_s, v_s) = (\alpha_s, -\sigma_s)$. Then $\Delta_s = \alpha_s^2 + \sigma_s^2$ and

$$a_s = \frac{\alpha_s}{\alpha_s^2 + \sigma_s^2}, \quad b_s = -\frac{\sigma_s}{\alpha_s^2 + \sigma_s^2}, \qquad \Rightarrow \quad \hat{\mathbf{x}}_0^{(s)} = \frac{\alpha_s}{\alpha_s^2 + \sigma_s^2} \mathbf{x}_s - \frac{\sigma_s}{\alpha_s^2 + \sigma_s^2} \psi^*(\mathbf{x}_s, s).$$

4. **Score-prediction (conditional score).** For the conditional Gaussian $q(\mathbf{x}_s \mid \mathbf{x}_0) = \mathcal{N}(\alpha_s \mathbf{x}_0, \sigma_s^2 \mathbf{I})$,

$$\nabla_{\mathbf{x}_s} \log q(\mathbf{x}_s \mid \mathbf{x}_0) = -\frac{\mathbf{x}_s - \alpha_s \mathbf{x}_0}{\sigma_s^2} = -\frac{1}{\sigma_s} \boldsymbol{\epsilon}.$$

Hence choosing $\psi_0 = \nabla_{\mathbf{x}_s} \log q(\mathbf{x}_s \mid \mathbf{x}_0)$ corresponds to $(u_s, v_s) = (-1/\sigma_s, 0)$. Then $\Delta_s = -\alpha_s/\sigma_s$ and the reconstruction coefficients are

$$a_s = \frac{1}{\alpha_s}, \quad b_s = \frac{\sigma_s^2}{\alpha_s}, \qquad \Rightarrow \quad \hat{\mathbf{x}}_0^{(s)} = \frac{1}{\alpha_s} \mathbf{x}_s + \frac{\sigma_s^2}{\alpha_s} \psi^*(\mathbf{x}_s, s).$$

5. **Score-prediction (marginal score).** Define the marginal distribution

$$q_s(\mathbf{x}_s) = \int q(\mathbf{x}_s \mid \mathbf{x}_0) \, p_{\text{data}}(\mathbf{x}_0) \, d\mathbf{x}_0.$$

By the Gaussian score identity,

$$\nabla_{\mathbf{x}_s} \log q_s(\mathbf{x}_s) = \mathbb{E}_{\mathbf{x}_0 \mid \mathbf{x}_s}[\nabla_{\mathbf{x}_s} \log q(\mathbf{x}_s \mid \mathbf{x}_0)] = -\frac{\mathbf{x}_s - \alpha_s \mathbb{E}[\mathbf{x}_0 \mid \mathbf{x}_s]}{\sigma_s^2}.$$

Table A.1: Unified view of common parameterizations. Each training target $\psi_0$ corresponds to a conditional regression, and the clean data $\mathbf{x}_0$ is exactly reconstructed via equation A.12.

| Prediction mode | Target $\psi_0(\mathbf{x}_0, \epsilon, s)$ | Reconstruction $\hat{\mathbf{x}}_0^{(s)}$ |
|---|---|---|
| $\epsilon$-prediction | $\epsilon$ | $\hat{\mathbf{x}}_0^{(s)} = \frac{1}{\alpha_s}\mathbf{x}_s - \frac{\sigma_s}{\alpha_s}\psi^*(\mathbf{x}_s, s)$ |
| $\mathbf{x_0}$-prediction | $\mathbf{x}_0$ | $\hat{\mathbf{x}}_0^{(s)} = \psi^*(\mathbf{x}_s, s)$ |
| $v$-prediction | $v = \alpha_s\epsilon - \sigma_s\mathbf{x}_0$ | $\hat{\mathbf{x}}_0^{(s)} = \frac{\alpha_s}{\alpha_s^2+\sigma_s^2}\mathbf{x}_s - \frac{\sigma_s}{\alpha_s^2+\sigma_s^2}\psi^*(\mathbf{x}_s, s)$ |
| $s$-prediction | $\nabla_{\mathbf{x}_s}\log q_s(\mathbf{x}_s) = -\frac{1}{\sigma_s}(\mathbf{x}_s - \frac{\alpha_s}{\alpha_s^2+\sigma_s^2}\mathbf{x}_0)$ | $\hat{\mathbf{x}}_0^{(s)} = \frac{1}{\alpha_s}\mathbf{x}_s + \frac{\sigma_s^2}{\alpha_s}\psi^*(\mathbf{x}_s, s)$ |
| Flow matching | $\epsilon - \mathbf{x}_0$ | $\hat{\mathbf{x}}_0^{(s)} = \mathbf{x}_s - s\,\psi^*(\mathbf{x}_s, s)$ |

Rearranging yields the posterior mean in closed form:

$$\mathbb{E}[\mathbf{x}_0 \mid \mathbf{x}_s] = \frac{1}{\alpha_s}\,\mathbf{x}_s \;+\; \frac{\sigma_s^2}{\alpha_s}\,\nabla_{\mathbf{x}_s}\log q_s(\mathbf{x}_s). \tag{A.13}$$

If the network is trained by DSM with squared loss so that $\psi^*(\mathbf{x}_s, s) = \nabla_{\mathbf{x}_s}\log q_s(\mathbf{x}_s)$, then equation A.13 gives the same reconstruction rule as in the conditional case:

$$\hat{\mathbf{x}}_0^{(s)} = \frac{1}{\alpha_s}\,\mathbf{x}_s + \frac{\sigma_s^2}{\alpha_s}\,\psi^*(\mathbf{x}_s, s),$$

with coefficients $(a_s, b_s) = (1/\alpha_s, \; \sigma_s^2/\alpha_s)$.

**Remark.** The coefficients depend only on the forward schedule $(\alpha_s, \sigma_s)$; the data distribution $p_{\text{data}}$ appears solely through the value of the marginal score $\nabla_{\mathbf{x}_s}\log q_s(\mathbf{x}_s)$ that the network estimates.

6. **Flow matching.** For the linear path $\mathbf{x}_s = (1-s)\mathbf{x}_0 + s\,\epsilon$ we have $\alpha_s = 1 - s$, $\sigma_s = s$. A common target is $\psi_0 = \epsilon - \mathbf{x}_0$, i.e. $(u_s, v_s) = (1, -1)$, so $\Delta_s = \alpha_s + \sigma_s = 1$ and

$$a_s = 1, \quad b_s = -\sigma_s = -s, \qquad \Rightarrow \quad \hat{\mathbf{x}}_0^{(s)} = \mathbf{x}_s - s\,\psi^*(\mathbf{x}_s, s).$$

Other normalizations (e.g. scaling $\psi_0$ by $(1-s)$) yield the corresponding $(a_s, b_s)$ via equation A.11 with the modified $(u_s, v_s)$.

In all cases, equation A.12 shows that training with any admissible $\psi_0$ recovers the same posterior mean of $\mathbf{x}_0$ from $(\mathbf{x}_s, s)$ up to a deterministic linear readout $(a_s, b_s)$ determined solely by $(\alpha_s, \sigma_s)$ and the chosen $(u_s, v_s)$. Hence the parameterizations are equivalent in expressive power: they differ only in the regression target and in the reconstruction coefficients $(a_s, b_s)$. $\qquad\square$

### A.3 Why Second-Order Differences are Optimal

We now provide a mathematical justification, entirely driven by the setting of Section A.2 and Section A, for why second-order differences are (i) *necessary and advantageous*, (ii) *sufficient and information-theoretically optimal*, and (iii) why *higher-order schemes degrade in practice*. Throughout, $t$ indexes discrete steps, while $s \in [0,1]$ denotes continuous time.

#### A.3.1 Why Second-Order Differences Are Good

**Setting.** Let $(\Omega, \mathcal{F}, \mathrm{Pr})$ be a probability space. A data sample is $x_0 \in L^2(\Omega; \mathbb{R}^d)$ with distribution $p_{\mathrm{data}}$. Let $\epsilon \sim \mathcal{N}(\mathbf{0}, \mathbf{I}_d)$ be independent of $x_0$. Fix $C^2$ schedules $\alpha_s, \sigma_s : [0,1] \to \mathbb{R}$ and define the forward latent

$$x_s = \alpha_s x_0 + \sigma_s \epsilon, \qquad s \in [0,1].$$

Let $\psi_\theta : \mathbb{R}^d \times [0,1] \to \mathbb{R}^d$ be a trained network, twice continuously differentiable in both arguments and of at most linear growth so that $\sup_{s \in [0,1]} \mathbb{E}\|\psi_\theta(x_s, s)\|^2 < \infty$. For discrete steps $t \in \mathbb{Z}$ we write $\psi_t := \psi_\theta(\mathbf{x}_{s_t}, s_t)$ with a locally uniform grid $s_{t\pm 1} = s_t \pm \Delta$.

> **Theorem A.4** (Signal–noise decomposition and invariance). *There exists a unique $\phi \in C^2([0,1]; \mathbb{R}^d)$ and a zero-mean perturbation $\eta_t$ with $\sup_t \mathbb{E}\|\eta_t\|^2 < \infty$ such that*
>
> $$\psi_t = \phi(s_t) + \eta_t, \qquad \mathbb{E}[\eta_t \mid s_t] = 0.$$
>
> *Hence the statement holds uniformly for the usual parameterizations ($\epsilon$, $\mathbf{x_0}$, $v$, score, and flow), which are related by deterministic affine readouts in $s$.*

*Proof.* Let

$$\mathcal{G} := \{\varphi(s_t) : \varphi : [0,1] \to \mathbb{R}^d, \ \mathbb{E}\|\varphi(s_t)\|^2 < \infty\} \subset L^2(\Omega; \mathbb{R}^d)$$

It is a closed subspace. The orthogonal projection of $\psi_t$ onto $\mathcal{G}$ is $\phi(s_t) := \mathbb{E}[\psi_t \mid s_t]$, and $\eta_t := \psi_t - \phi(s_t)$ satisfies $\mathbb{E}[\eta_t \mid s_t] = 0$.

Uniqueness follows from the uniqueness of Hilbert projections.

Write $F(s, \mathbf{x}_0, \epsilon) := \psi_\theta(\alpha_s \mathbf{x}_0 + \sigma_s \epsilon, s)$, so $\phi(s) = \mathbb{E}[F(s, \mathbf{x}_0, \epsilon)]$. Since $\psi_\theta \in C^2$ and $\alpha_s, \sigma_s \in C^2$, chain rule yields

$$\partial_s F = \partial_s \psi_\theta(\mathbf{x}_s, s) + \nabla_x \psi_\theta(\mathbf{x}_s, s)(\alpha'_s \mathbf{x}_0 + \sigma'_s \epsilon),$$

$$\partial_s^2 F = \partial_s^2 \psi_\theta + 2\, \partial_s \nabla_x \psi_\theta\, (\alpha'_s \mathbf{x}_0 + \sigma'_s \epsilon) + \nabla_x \psi_\theta\, (\alpha''_s \mathbf{x}_0 + \sigma''_s \epsilon) + (\alpha'_s \mathbf{x}_0 + \sigma'_s \epsilon)^\top \nabla_x^2 \psi_\theta\, (\alpha'_s \mathbf{x}_0 + \sigma'_s \epsilon),$$

all evaluated at $(\mathbf{x}_s, s)$. By the polynomial growth assumption and $\mathbf{x}_0, \epsilon \in L^2$, $\partial_s F$ and $\partial_s^2 F$ are dominated by integrable envelopes. Thus, by dominated convergence (interchange of limit and expectation),

$$\phi'(s) = \mathbb{E}[\partial_s F], \qquad \phi''(s) = \mathbb{E}[\partial_s^2 F],$$

so $\phi \in C^2([0,1])$.

Bounded variance follows from $\mathbb{E}\|\eta_t\|^2 \leq 2 \sup_s \mathbb{E}\|\psi_\theta(x_s, s)\|^2 + 2 \sup_s \|\phi(s)\|^2 < \infty$. $\qquad\square$

Theorem A.4 justifies the local model $\psi_u = \phi(u) + \eta_u$ with a smooth deterministic trend $\phi$ and a zero-mean perturbation $\eta_u$. We now study the task of *reconstructing a skipped state* $\psi_{t-\Delta}$ from the two most recent computed states $\{\psi_t, \psi_{t+\Delta}\}$.

> **Theorem A.5** (Second-order backward extrapolation is BLUE and second-order accurate). *Assume the decomposition in Theorem A.4 and a locally uniform grid $s_{t\pm 1} = s_t \pm \Delta$. Consider linear estimators $\widehat{\psi}_{t-\Delta} = a\, \psi_t + b\, \psi_{t+\Delta}$ that are unbiased for all affine trends $\phi(u) = \beta_0 + \beta_1 u$. Then $a = 2$, $b = -1$ is the unique unbiased choice, and under homoscedastic uncorrelated perturbations $\mathrm{Var}([\eta_t, \eta_{t+\Delta}]^\top) = \sigma^2 I_2$ it minimizes the variance among all unbiased linear estimators. Moreover,*
>
> $$\mathbb{E}[\psi_{t-\Delta} - (2\psi_t - \psi_{t+\Delta})] = \Delta^2\, \phi''(s_t) + o(\Delta^2),$$

*so the bias is $O(\Delta^2)$ for $\phi \in C^2$.*

*Proof.* Write $y_t := \psi_t$, $y_{t+\Delta} := \psi_{t+\Delta}$ and stack $y = \begin{bmatrix} y_t \\ y_{t+\Delta} \end{bmatrix}$. Under an affine trend, $y = X\beta + \eta$ with

$$X = \begin{bmatrix} 1 & s_t \\ 1 & s_t + \Delta \end{bmatrix}, \quad \beta = \begin{bmatrix} \beta_0 \\ \beta_1 \end{bmatrix}, \quad \eta = \begin{bmatrix} \eta_t \\ \eta_{t+\Delta} \end{bmatrix}.$$

The target is $\theta := \phi(s_t - \Delta) = c^\top \beta$ with $c = \begin{bmatrix} 1 \\ s_t - \Delta \end{bmatrix}$. A linear estimator $w^\top y$ is unbiased for all affine $\phi$ iff $w^\top X = c^\top$, i.e., $X^\top w = c$. Solving yields $w = (2, -1)^\top$ and hence $\widehat{\psi}_{t-\Delta} = 2\psi_t - \psi_{t+\Delta}$.

For variance optimality with $\mathrm{Var}(\eta) = \sigma^2 I_2$, Gauss–Markov gives

$$w^\star = \arg\min_{w:\, X^\top w = c} \mathrm{Var}(w^\top y) = \arg\min \sigma^2 \|w\|_2^2 \quad \Rightarrow \quad w^\star = X(X^\top X)^{-1} c = (2, -1)^\top.$$

Thus $2\psi_t - \psi_{t+\Delta}$ is the BLUE.

For bias, expand $\phi$ at $s_t$:

$$\phi(s_t \pm \Delta) = \phi(s_t) \pm \Delta\, \phi'(s_t) + \tfrac{\Delta^2}{2} \phi''(s_t) + o(\Delta^2).$$

Hence

$$\phi(s_t - \Delta) - \big(2\phi(s_t) - \phi(s_t + \Delta)\big) = \Delta^2\, \phi''(s_t) + o(\Delta^2).$$

Adding the zero-mean perturbations on both sides preserves the expansion in mean, proving the stated local truncation error. $\square$

*Remark* A.6 (On conditioning in Theorem A.4: $\phi(s)$ is a population-level trend). Let $\mathbf{x_s} = \alpha_s \mathbf{x_0} + \sigma_s \boldsymbol{\epsilon}$ with $\mathbf{x_0} \sim p_{\text{data}}$, $\boldsymbol{\epsilon} \sim \mathcal{N}(0, I)$, and assume $s$ is independent of $(\mathbf{x_0}, \boldsymbol{\epsilon})$. For a target $\psi_t = \psi_0(\mathbf{x_0}, \boldsymbol{\epsilon}, s_t)$, Theorem A.4 defines the signal part by

$$\phi(s) := \mathbb{E}[\psi_t \,|\, s_t = s].$$

Importantly, the conditioning is *only* on the time index $s$ (not on the realization $\mathbf{x_s}$), hence $\phi$ is a deterministic function of $s$. By the tower property,

$$\phi(s) = \mathbb{E}\big[\psi_0(\mathbf{x_0}, \boldsymbol{\epsilon}, s) \,\big|\, s\big] = \mathbb{E}\big[\,\mathbb{E}\big[\psi_0(\mathbf{x_0}, \boldsymbol{\epsilon}, s) \,\big|\, \mathbf{x_s}, s\big] \,\big|\, s\big],$$

so $\phi(s)$ is obtained by first taking the posterior mean given $(\mathbf{x_s}, s)$ and then marginalizing over $\mathbf{x_s} \sim q_s$. This is fundamentally different from *posterior reconstruction* (e.g., $\hat{\mathbf{x}}_0(\mathbf{x_s}, s) := \mathbb{E}[\mathbf{x_0} \,|\, \mathbf{x_s}, s]$), which depends on the particular observation $\mathbf{x_s}$.

For the usual parameterizations we get closed forms. We use that $\mathbb{E}[\boldsymbol{\epsilon} \,|\, s] = \mathbb{E}[\boldsymbol{\epsilon}] = 0$ and $\mathbb{E}[\mathbf{x_0} \,|\, s] = \mathbb{E}[\mathbf{x_0}]$ by independence.

- $\boldsymbol{\epsilon}$-**prediction:** $\psi_0(\mathbf{x_0}, \boldsymbol{\epsilon}, s) = \boldsymbol{\epsilon}$. Then $\phi(s) = \mathbb{E}[\boldsymbol{\epsilon} \,|\, s] = 0$.

- $\mathbf{x_0}$-**prediction:** $\psi_0(\mathbf{x_0}, \boldsymbol{\epsilon}, s) = \mathbf{x_0}$. Then $\phi(s) = \mathbb{E}[\mathbf{x_0} \,|\, s] = \mathbb{E}[\mathbf{x_0}]$.

- $v$-**prediction:** $\psi_0(\mathbf{x_0}, \boldsymbol{\epsilon}, s) = \alpha_s \boldsymbol{\epsilon} - \sigma_s \mathbf{x_0}$. Hence $\phi(s) = \alpha_s \mathbb{E}[\boldsymbol{\epsilon} \,|\, s] - \sigma_s \mathbb{E}[\mathbf{x_0} \,|\, s] = -\sigma_s\, \mathbb{E}[\mathbf{x_0}]$.

- **Score-prediction:** $\psi_0(\mathbf{x_0}, \boldsymbol{\epsilon}, s) = \nabla_{\mathbf{x_s}} \log q_s(\mathbf{x_s})$, so $\phi(s) = \mathbb{E}_{\mathbf{x_s} \sim q_s}[\nabla \log q_s(\mathbf{x_s})] = 0$.

- **Flow-prediction:** $\psi_0(\mathbf{x_0}, \boldsymbol{\epsilon}, s) = \frac{d}{ds}\mathbf{x_s} = \alpha_s' \mathbf{x_0} + \sigma_s' \boldsymbol{\epsilon}$, so $\phi(s) = \alpha_s' \mathbb{E}[\mathbf{x_0} \,|\, s] + \sigma_s' \mathbb{E}[\boldsymbol{\epsilon} \,|\, s] = \alpha_s' \mathbb{E}[\mathbf{x_0}]$.

Equivalently, whenever a parameterization is an affine readout $\psi_0(\mathbf{x_0}, \boldsymbol{\epsilon}, s) = a(s)\, \boldsymbol{\epsilon} + b(s)\, \mathbf{x_0} + c(s)$, we have the general identity

$$\phi(s) = b(s)\, \mathbb{E}[\mathbf{x_0}] + c(s),$$

since $\mathbb{E}[\boldsymbol{\epsilon}] = 0$ and $s \perp (\mathbf{x_0}, \boldsymbol{\epsilon})$.

Thus assuming that $\phi(s)$ is an afine trends is reasonable.

Table A.2: Summary of different parameterizations.

| Parameterization | $\phi(s) = \mathbb{E}[\psi_t \mid s_t = s]$ |
|---|---|
| $\epsilon$-prediction | $0$ |
| $\mathbf{x_0}$-prediction | $\mathbb{E}[\mathbf{x_0}]$ |
| $v$-prediction | $-\sigma_s\,\mathbb{E}[\mathbf{x_0}]$ |
| Score-prediction | $0$ |
| Flow-prediction | $\alpha'_s\,\mathbb{E}[\mathbf{x_0}]$ |

**Proposition A.7** (No two-point linear estimator beats the $O(\Delta^2)$ order). *Let $\widehat{\psi}_{t-\Delta} = a\,\psi_t + b\,\psi_{t+\Delta}$ be any estimator that is unbiased for all affine $\phi$. Then for every such choice,*

$$\psi_{t-\Delta} - \widehat{\psi}_{t-\Delta} = K(a,b)\,\Delta^2\,\phi''(s_t) + o(\Delta^2) \quad \text{with } K(a,b) \neq 0,$$

*so the order $O(\Delta^2)$ cannot be improved using only $\{\psi_t, \psi_{t+\Delta}\}$.*

*Proof.* Unbiasedness for all affine $\phi$ enforces the constraints $a + b = 1$ and $as_t + b(s_t + \Delta) = s_t - \Delta$, which have the unique solution $a = 2$, $b = -1$. For any $C^2$ trend, Taylor's theorem with remainder gives the error coefficient $K(2, -1) = 1$. If one relaxed unbiasedness, matching constants and linears is still necessary to avoid $O(1)$ or $O(\Delta)$ bias uniformly in $\phi$; with only two samples, the highest degree one can reproduce is 1, hence the Peano kernel argument yields an $O(\Delta^2)$ remainder with a nonzero coefficient for some $\phi''$. $\square$

**Corollary A.8** (Curvature observability). *The second difference $\Delta^{(2)}\psi_t := \psi_{t+\Delta} - 2\psi_t + \psi_{t-\Delta}$ cancels any affine trend and isolates curvature:*

$$\Delta^{(2)}\phi(s_t) = \Delta^2\,\phi''(s_t) + o(\Delta^2).$$

*Thus the BLUE extrapolator $2\psi_t - \psi_{t+\Delta}$ explicitly exploits $\Delta^{(2)}$ to reconstruct the skipped state $\psi_{t-\Delta}$ while being insensitive to large affine drifts in $\phi$.*

*Remark* A.9 (Correlated perturbations and generalized least squares). If $\mathrm{Var}(\eta) = \Sigma \succ 0$ (not necessarily diagonal), the BLUE weights become

$$(w^\star)^\top = c^\top \left(X^\top \Sigma^{-1} X\right)^{-1} X^\top \Sigma^{-1}.$$

When $\Sigma = \sigma^2 I_2$ this reduces to $(2, -1)$.

**Takeaway.** Under the mild, parameterization-invariant decomposition of Theorem A.4, the backward second-order rule

$$\widehat{\psi}_{t-\Delta} = 2\psi_t - \psi_{t+\Delta}$$

is simultaneously (i) unbiased for all affine trends, (ii) variance-optimal among all linear unbiased two-point estimators, (iii) second-order accurate with bias $O(\Delta^2)$, and (iv) curvature-aware through $\Delta^{(2)}$. None of these guarantees is achievable with zeroth- or first-order reuse from $\{\psi_t, \psi_{t+\Delta}\}$ alone.

From the next section onward, we omit the perturbation terms $\eta$ and focus solely on the underlying trend $\phi$, as the $(2, -1)$ rule has already been shown to be the unique BLUE. Any additive zero-mean noise merely inflates the estimation variance but cannot reduce the inherent bias floor.

### A.3.2 WHY SECOND-ORDER IS SUFFICIENT

**Setup.** Let the unknown target function $\phi : [0, 1] \to \mathbb{R}^d$ belong to the class

$$\mathcal{F}(M_2) = \left\{\phi \in C^2([0,1]; \mathbb{R}^d) : \sup_{s \in [0,1]} \|\phi''(s)\| \leq M_2\right\}.$$

At discrete step $t$, we only have access to the model evaluations

$$\psi_t = \phi(s_t), \qquad \psi_{t+\Delta} = \phi(s_{t+\Delta}),$$

with $s_{t\pm\Delta} = s_t \pm \Delta$. The goal is to estimate $\phi(s_{t-\Delta})$ based on these observations. Denote a generic estimator by

$$\widehat{\psi}_{t-\Delta} = \mathsf{Alg}(\psi_t, \psi_{t+\Delta}).$$

We show that under only $C^2$ regularity, every estimator incurs worst-case bias of order $\Delta^2$, while the standard second-order extrapolation

$$\widehat{\psi}_{t-\Delta}^{(2)} = 2\,\psi_t - \psi_{t+\Delta} \tag{A.14}$$

achieves this rate. Thus, second-order is minimax optimal.

**Lower Bound via Two-Point Method.**

**Theorem A.10** (Two-point lower bound). *For any estimator $\widehat{\psi}_{t-\Delta}$ depending only on $\{\psi_t, \psi_{t+\Delta}\}$, there exist $\phi_\pm \in \mathcal{F}(M_2)$ such that*

$$\phi_\pm(s_t) = \phi_\pm(s_{t+\Delta}) = 0, \qquad \big\|\phi_+(s_{t-\Delta}) - \phi_-(s_{t-\Delta})\big\| \geq c\,M_2\,\Delta^2,$$

*for some absolute constant $c > 0$. Consequently,*

$$\inf_{\widehat{\psi}_{t-\Delta}} \sup_{\phi \in \mathcal{F}(M_2)} \big\|\widehat{\psi}_{t-\Delta} - \phi(s_{t-\Delta})\big\| = \Omega(M_2\Delta^2).$$

*Proof.* Let $u = (s - s_t)/\Delta$ and consider the quadratic Lagrange basis polynomial

$$p(u) = \tfrac{1}{2}u(u-1), \qquad p(0) = p(1) = 0,\ p(-1) = 1,\ p''(u) \equiv 1.$$

Define $g(s) = M_2\Delta^2\,p\big((s-s_t)/\Delta\big)$ and set $\phi_\pm = \pm g$. Then $\phi_\pm \in \mathcal{F}(M_2)$, agree at $s_t$ and $s_{t+\Delta}$, but differ by

$$\|\phi_+(s_{t-\Delta}) - \phi_-(s_{t-\Delta})\| = 2M_2\Delta^2.$$

Since the data are indistinguishable, Le Cam's two-point method (LeCam, 1973) implies any estimator incurs at least half this separation on one of the two instances, yielding the stated $\Omega(M_2\Delta^2)$ bound. $\square$

**Transition.** Theorem 1 shows that $\Delta^2$ bias is an information-theoretic lower bound. Next we show that the second-order extrapolation equation A.14 matches this rate.

**Upper Bound via Second-Order Extrapolation.**

**Theorem A.11** (Achievability). *For any $\phi \in \mathcal{F}(M_2)$,*

$$\big\|\phi(s_{t-\Delta}) - (2\phi(s_t) - \phi(s_{t+\Delta}))\big\| \leq M_2\,\Delta^2 = \mathcal{O}(M_2\Delta^2).$$

*Proof.* By Taylor's theorem, for some $\theta_1, \theta_2 \in (0,1)$,

$$\phi(s_{t+\Delta}) = \phi(s_t) + \phi'(s_t)\Delta + \tfrac{1}{2}\,\phi''(s_t + \theta_1\Delta)\Delta^2,$$
$$\phi(s_{t-\Delta}) = \phi(s_t) - \phi'(s_t)\Delta + \tfrac{1}{2}\,\phi''(s_t - \theta_2\Delta)\Delta^2.$$

Subtracting yields

$$\phi(s_{t-\Delta}) - \big(2\phi(s_t) - \phi(s_{t+\Delta})\big) = \tfrac{1}{2}\big[\phi''(s_t - \theta_2\Delta) + \phi''(s_t + \theta_1\Delta)\big]\Delta^2,$$

whose norm is bounded by $M_2\Delta^2$. $\square$

**Corollary A.12** (Minimax rate). *Combining Theorems 1 and 2, the minimax rate for estimating $\phi(s_{t-\Delta})$ under $C^2$ regularity is $\Delta^2$, achieved by the second-order extrapolation equation A.14.*

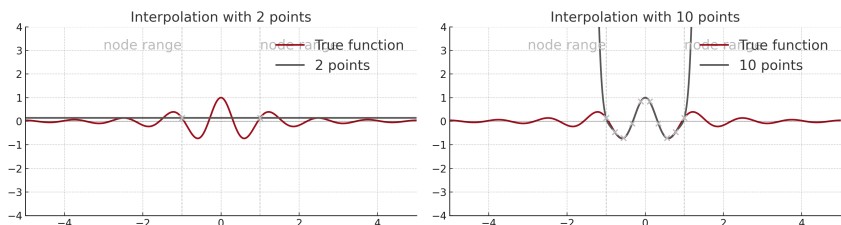

Figure A.1: Illustration of the Runge phenomenon. Left: polynomial interpolation with only 2 nodes. Right: interpolation with 10 nodes, which exhibits severe oscillations and divergence near the boundary.

**Variance and BLUE.** Suppose further that observations are corrupted by i.i.d. noise:

$$\psi_t = \phi(s_t) + \eta_t, \qquad \psi_{t+\Delta} = \phi(s_{t+\Delta}) + \eta_{t+\Delta}, \qquad \mathbb{E}[\eta_t] = 0, \ \mathrm{Var}(\eta_t) = \sigma^2.$$

Consider linear unbiased estimators $\widehat{\psi}_{t-\Delta} = a\psi_t + b\psi_{t+\Delta}$. Unbiasedness for all affine $\phi(s)$ requires

$$\begin{bmatrix} 1 & 1 \\ s_t & s_{t+\Delta} \end{bmatrix} \begin{bmatrix} a \\ b \end{bmatrix} = \begin{bmatrix} 1 \\ s_{t-\Delta} \end{bmatrix},$$

whose unique solution is $(a, b) = (2, -1)$. The variance is then

$$\mathrm{Var}(\widehat{\psi}_{t-\Delta}) = (a^2 + b^2)\sigma^2 = 5\sigma^2.$$

Thus equation A.14 is the unique best linear unbiased estimator (BLUE).

**Complexity Perspective.** Three points uniquely determine a quadratic interpolant, achieving order-$\Delta^2$ bias. Adding more points to fit higher-degree polynomials cannot improve the minimax rate, since functions in $\mathcal{F}(M_2)$ need not possess bounded higher derivatives. In fact, the Lebesgue constant

$$\Lambda_n = \max_s \sum_{j=0}^{n} |\ell_j(s)|$$

of polynomial interpolation typically grows with the number of nodes, degrading stability. Hence "more points" only increase constants without reducing the minimax order. Moreover, since our sampling points are nearly uniform, the resulting polynomial interpolant is susceptible to Runge's phenomenon (Runge et al., 1901) (see Figure A.1), which may further degrade stability (see Theorem A.16 for more details).

**Conclusion.** Under scarce fresh evaluations and only $C^2$ regularity (the weakest assumption justified by the forward process, cf. Appendix A.1), second-order extrapolation equation A.14 is *information-theoretically optimal*: it matches the $\Delta^2$ minimax lower bound and is BLUE among linear unbiased estimators. Thus it lies on the Pareto frontier (Pareto, 1919) of the bias–variance tradeoff, and additional points cannot improve the minimax rate while often worsening numerical stability.

### A.3.3 WHY FIRST-ORDER REUSE IS INSUFFICIENT

We quantify the best possible accuracy if one only reuses a *single* model evaluation (e.g., $\widehat{\psi}_{t-\Delta} = \psi_t$ or $\widehat{\psi}_{t-\Delta} = \psi_{t+\Delta}$). Let

$$\mathcal{F}_1(M_1) = \left\{ \phi \in C^1([0,1]; \mathbb{R}^d) : \sup_{s\in[0,1]} \|\phi'(s)\| \leq M_1 \right\}.$$

Since $\phi \in C^2([0,1])$ on a compact interval implies $\phi' \in C^1$ and hence bounded, the class $\mathcal{F}_1(M_1)$ is compatible with the $C^2$ setting of §A.3.2.

**Information-theoretic lower bound for one-point estimators.** Consider any estimator that depends on a single point,

$$\widehat{\psi}_{t-\Delta} \;=\; \mathsf{Alg}(\psi_\tau), \qquad \tau \in \{t,\; t+\Delta\},$$

with $\psi_u = \phi(s_u)$ and $s_{t\pm\Delta} = s_t \pm \Delta$.

> **Theorem A.13** (One-point minimax lower bound)**.** *For any measurable estimator* $\widehat{\psi}_{t-\Delta} = \mathsf{Alg}(\psi_\tau)$ *using only one of* $\psi_t, \psi_{t+\Delta}$,
>
> $$\inf_{\widehat{\psi}_{t-\Delta}} \sup_{\phi \in \mathcal{F}_1(M_1)} \big\|\widehat{\psi}_{t-\Delta} - \phi(s_{t-\Delta})\big\| \;=\; \Omega(M_1 \Delta).$$

*Proof.* WLOG take $\tau = t$ (the case $\tau = t+\Delta$ is analogous). Define two affine trends

$$\phi_\pm(s) \;=\; \pm M_1\,(s - s_t).$$

Then $\phi_\pm \in \mathcal{F}_1(M_1)$, and they *agree* at $s_t$: $\phi_+(s_t) = \phi_-(s_t) = 0$, hence the observation $\psi_t$ is identical in both cases. But at the target location,

$$\big\|\phi_+(s_{t-\Delta}) - \phi_-(s_{t-\Delta})\big\| = \big\|+M_1(-\Delta) - (-M_1(-\Delta))\big\| = 2M_1\Delta.$$

Since the data are indistinguishable, Le Cam's two-point method (LeCam, 1973) implies any estimator incurs at least half this separation on one of the two instances, yielding the stated $\Omega(M_1\Delta)$ bound. $\qquad\square$

**Achievability with naive reuse.** The trivial reuse $\widehat{\psi}_{t-\Delta} = \psi_t$ attains this rate.

> **Proposition A.14** (First-order bias upper bound)**.** *For any* $\phi \in \mathcal{F}_1(M_1)$,
>
> $$\big\|\phi(s_{t-\Delta}) - \psi_t\big\| = \big\|\phi(s_{t-\Delta}) - \phi(s_t)\big\| \;\leq\; M_1\,\Delta.$$
>
> *Thus the one-point minimax rate is* $\Theta(M_1\Delta)$.

*Proof.* By the mean-value theorem, $\phi(s_{t-\Delta}) - \phi(s_t) = -\Delta\,\phi'(s_t - \theta\Delta)$ for some $\theta \in (0,1)$, so the norm is $\leq M_1\Delta$. $\qquad\square$

**Consequences (vs. second-order).**

- **Bias order:** Any one-point scheme is *at best* first-order accurate (bias $\Theta(\Delta)$), while the two-point second-order extrapolation $2\psi_t - \psi_{t+\Delta}$ is *second-order* (bias $\Theta(\Delta^2)$), cf. Section A.3.2.
- **Linear-unbiased restriction:** If we additionally require linear unbiasedness for constants (natural for reuse), $\widehat{\psi}_{t-\Delta} = a\,\psi_t$ forces $a = 1$, so

  $$\phi(s_{t-\Delta}) - \widehat{\psi}_{t-\Delta} = \phi(s_{t-\Delta}) - \phi(s_t) = -\Delta\,\phi'(s_t) + \tfrac{\Delta^2}{2}\phi''(s_t) + o(\Delta^2),$$

  whose leading term is generically $O(\Delta)$ and cannot be removed without using *two* points to cancel the linear drift.

**Takeaway.** Any estimator that "just reuses" a single evaluation is *information-theoretically* limited to an $O(\Delta)$ bias (Theorem A.13), which the trivial reuse $\widehat{\psi}_{t-\Delta} = \psi_t$ already attains (Proposition A.14). In contrast, the two-point second-order rule cancels the linear drift and reaches the $\Theta(\Delta^2)$ minimax rate under $C^2$ regularity (Theorem A.10 and Theorem A.11). Hence first-order reuse is *necessarily suboptimal*.

### A.3.4 WHY HIGHER-ORDER EXTRAPOLATION IS DETRIMENTAL

We now establish, in a formal manner, why schemes of order $k > 2$ are not advantageous under the $C^2$ regularity available in diffusion ODE sampling. Despite the apparent flexibility of higher-order stencils, they suffer from four fundamental drawbacks: exponential noise amplification, curvature mixing, collapse of stability domain, and interpolation sensitivity.

**Theorem A.15** (Exponential noise amplification). *Let $w^{(k)} \in \mathbb{R}^{k+1}$ be the Lagrange weights for extrapolating $\psi_{-1}$ from $\{\psi_0, \ldots, \psi_k\}$. Then*

$$\|w^{(k)}\|_2^2 = \binom{2k+2}{k+1} \sim \frac{4^{k+1}}{\sqrt{\pi(k+1)}},$$

*and hence the variance of the extrapolate obeys* $\mathrm{Var}(\widehat{\psi}_{-1}^{(k)}) = \Omega(4^k \sigma^2)$.

*Proof.* For equispaced nodes $\{0, 1, \ldots, k\}$ the $j$-th Lagrange basis reads

$$l_j(x) = \prod_{\substack{m=0 \\ m \neq j}}^{k} \frac{x - m}{j - m}.$$

Evaluating at $x = -1$ gives

$$w_j^{(k)} = l_j(-1) = \frac{\prod_{m=0, m \neq j}^{k} (-1 - m)}{\prod_{m=0, m \neq j}^{k} (j - m)}.$$

The numerator simplifies as

$$\prod_{\substack{m=0, m \neq j}}^{k} (-1 - m) = \frac{\prod_{m=0}^{k} (-1 - m)}{-1 - j} = \frac{(-1)^{k+1} (k+1)!}{-(j+1)} = (-1)^k \frac{(k+1)!}{j+1}.$$

The denominator splits into two products:

$$\prod_{\substack{m=0, m \neq j}}^{k} (j - m) = \left( \prod_{m=0}^{j-1} (j - m) \right) \left( \prod_{m=j+1}^{k} (j - m) \right) = j! \, (-1)^{k-j} (k - j)!.$$

Combining yields

$$w_j^{(k)} = (-1)^j \frac{(k+1)!}{(j+1)! \, (k-j)!} = (-1)^j \binom{k+1}{j+1}.$$

Therefore the squared $\ell_2$ norm is

$$\|w^{(k)}\|_2^2 = \sum_{j=0}^{k} \binom{k+1}{j+1}^2 = \sum_{m=1}^{k+1} \binom{k+1}{m}^2.$$

By the Chu–Vandermonde identity $\sum_{m=0}^{n} \binom{n}{m}^2 = \binom{2n}{n}$, we obtain

$$\|w^{(k)}\|_2^2 = \binom{2k+2}{k+1}.$$

Finally, Stirling's formula for the central binomial coefficient gives

$$\binom{2k+2}{k+1} \sim \frac{4^{k+1}}{\sqrt{\pi(k+1)}},$$

hence

$$\mathrm{Var}(\widehat{\psi}_{-1}^{(k)}) = \sigma^2 \|w^{(k)}\|_2^2 = \Omega(4^k \sigma^2),$$

as claimed. $\square$

**Theorem A.16** (Interpolation sensitivity). *Let $\Lambda_k$ be the Lebesgue constant of equispaced inter-polation using $k + 1$ nodes. Then $\Lambda_k$ grows exponentially in $k$, whereas for optimal Chebyshev nodes it grows only logarithmically. Hence, higher-order extrapolants with equispaced stencils are exponentially sensitive to perturbations or irregular steps.*

*Proof.* The exponential growth already appeared in Theorem A.15: the $\ell_2$ norm of the weights $\|w^{(k)}\|_2$ scales like $4^k$, so any perturbation in the data is magnified accordingly. This phenomenon is precisely quantified by the Lebesgue constant

$$\Lambda_k = \sup_x \sum_{j=0}^{k} |l_j(x)|,$$

which measures the operator norm of the interpolation map. Classical interpolation theory (Runge et al., 1901) shows that for equispaced nodes $\Lambda_k \sim c \, 2^k$ with $c > 0$. Since diffusion ODE sampling necessarily uses uniform timesteps, we inherit the exponential sensitivity of equispaced interpolation. $\square$

**Conclusion.** Together, these theorems show that higher-order extrapolation schemes amplify stochastic noise by $\Omega(4^k)$, destroy curvature alignment through alternating differences, shrink the admissible stability domain beyond usefulness, and become exponentially sensitive to small pertur-bations. Under the $C^2$ smoothness regime of diffusion processes, the minimax bias rate is $\Omega(\Delta^2)$, already attained by second-order schemes. Thus higher-order methods offer no bias improvement but introduce severe variance and stability costs. In realistic sampling budgets, second-order extrapolation is both necessary and sufficient for robust skipping in diffusion ODEs.

### A.4 WHY IS SECOND-ORDER DIFFERENCING THE ONLY POSSIBILITY UNDER AMBITIOUS SKIPPING?

We now make precise the consequence of enforcing uniform skipping with a speed-up factor of at least two. Under such uniform spacing, the pigeonhole principle implies that no two consecutive steps can both be fresh, and every three-step local window contains at most one fresh evaluation. As a result, the only minimally sufficient real-computation unit is formed by a single fresh value together with its deterministic difference against the next state, yielding a unique second-order difference rule.

**Theorem A.17** (Uniform $\geq 2\times$ skipping forbids consecutive fresh steps and yields a unique minimal tuple). *Let steps be $1, 2, \ldots, n$ on a uniform grid. Each step is either produced by a fresh network evaluation ("fresh") or by purely algebraic reuse of already computed values. Assume the skipping pattern is* uniform*: there exists an integer $r \geq 2$ and a residue class $c \in \{1, \ldots, r\}$ such that the fresh steps are exactly those indices*

$$\mathcal{F} = \{ i \in \{1, \ldots, n\} : i \equiv c \pmod{r} \},$$

*and every other step is obtained by algebraic reuse (no extra network calls). Then:*

1. *No two consecutive steps can both be fresh.*

2. *In any local window $\{t - 1, t, t + 1\}$, there is at most one fresh step.*

3. *If $\psi_{t-1}$ is skipped (to be reconstructed by reuse), the only minimally viable real-computation tuple is*

$$\left( \psi_t, \, \Delta\psi_t \right), \qquad \Delta\psi_t := \psi_t - \psi_{t+1},$$

*in the sense that $\psi_t$ is the unique fresh evaluation in the window and $\Delta\psi_t$ is a deter-ministic difference (trend) computable without an additional network call. No strictly smaller tuple can determine both level and local trend, and any tuple containing two fresh entries contradicts the uniform $r \geq 2$ spacing.*

*Proof.* By uniformity, any two fresh indices differ by a multiple of $r \geq 2$. Hence the gap between consecutive fresh steps is at least 2, so no two adjacent indices can both be fresh, proving (1).

Consequently, any three-consecutive-step window $\{t-1, t, t+1\}$ contains at most one fresh index, proving (2).

For (3), consider a window $\{t-1, t, t+1\}$ in which $\psi_{t-1}$ is skipped. By (2), at most one of these is fresh. If $t-1$ were fresh, no reconstruction would be needed; thus (w.l.o.g.) $t$ is the unique fresh index. Any additional quantity used to infer $\psi_{t-1}$ must be obtained without further network calls. Since $t+1$ cannot be fresh when $t$ is fresh under the uniform $r \geq 2$ spacing, reusing $\psi_{t+1}$ is deterministic. The first difference $\Delta\psi_t := \psi_t - \psi_{t+1}$ then supplies independent information about the local trend around $t$.

Minimality follows from identifiability: a single fresh value $\psi_t$ fixes only the local "level." Without at least one independent trend descriptor (e.g., $\Delta\psi_t$), $\psi_{t-1}$ is not determined in general (e.g., under an affine local model one needs both level and slope). Thus any tuple strictly smaller than $(\psi_t, \Delta\psi_t)$ is insufficient. Conversely, any tuple with two fresh entries violates the uniform gap $r \geq 2$. Hence $(\psi_t, \Delta\psi_t)$ is the unique minimally sufficient real-computation unit under uniform $\geq 2\times$ skipping. $\square$

### A.5 MULTI-STEP ERROR UNDER REUSE AND TWO-POINT EXTRAPOLATION

We study the bias and variance of three strategies for jumping left by $j$ grid points from an anchor at $s_t = t\Delta$ when only the two most recent observations $\{\psi_t, \psi_{t+1}\}$ are available. Assume a $d$-dimensional smooth ground-truth trajectory $\psi^* : [0,1] \to \mathbb{R}^d$ with $\psi^* \in C^4([0,1])$, and noisy observations

$$\psi_u = \psi^*(s_u) + \eta_u, \qquad \mathbb{E}[\eta_u] = 0, \quad \mathrm{Var}(\eta_u) = \sigma^2 I_d, \quad \eta_u \text{ uncorrelated across } u.$$

For any estimator $\widehat{\psi}_{t-j}$ of $\psi_{t-j}^* := \psi^*(s_{t-j})$, define its mean-bias and variance by

$$\mathrm{Bias}(\widehat{\psi}_{t-j}) := \mathbb{E}[\widehat{\psi}_{t-j}] - \psi_{t-j}^*, \qquad \mathrm{Var}(\widehat{\psi}_{t-j}) := \mathrm{Var}(\widehat{\psi}_{t-j}),$$

and the mean-squared error $\mathrm{MSE} = \|\mathrm{Bias}\|_2^2 + \mathrm{tr}\,\mathrm{Var}$. Below, all big-$O$ terms are uniform in $t$ and $j$ as $\Delta \to 0$; derivatives of $\psi^*$ are evaluated at $s_t$ unless stated otherwise.

**Three strategies.** (A) *Two-point linear extrapolation at a single anchor*: for any $j \geq 1$,

$$\widehat{\psi}_{t-j}^A := (j+1)\psi_t - j\psi_{t+1}.$$

(B) *Interval reuse*: first compute $\widehat{\psi}_{t-1} = 2\psi_t - \psi_{t+1}$, and then reuse every other step to the left, i.e. $\widehat{\psi}_s^B = \widehat{\psi}_{s+2}^B$ for $s \leq t-2$. Equivalently,

$$\widehat{\psi}_{t-j}^B = \begin{cases} \psi_t, & j \text{ even,} \\ 2\psi_t - \psi_{t+1}, & j \text{ odd.} \end{cases}$$

(C) *Pure reuse*: ignore $\psi_{t+1}$ and set $\widehat{\psi}_{t-j}^C := \psi_t$ for all $j \geq 1$.

**Lemma A.18** (Closed form and equivalence). *Strategy (A) is the unique sequence generated by the second-order recurrence $\widehat{\psi}_{t-(k+1)} = 2\widehat{\psi}_{t-k} - \widehat{\psi}_{t-(k-1)}$ with initial conditions $\widehat{\psi}_t = \psi_t$ and $\widehat{\psi}_{t-1} = 2\psi_t - \psi_{t+1}$. In particular, $\widehat{\psi}_{t-j}^A = (j+1)\psi_t - j\psi_{t+1}$ for all $j \geq 1$.*

*Proof.* Solve the linear homogeneous recurrence $x_{k+1} - 2x_k + x_{k-1} = 0$, whose general solution is $x_k = \alpha + \beta k$. With $\mathbf{x_0} = \psi_t$ and $x_1 = 2\psi_t - \psi_{t+1}$ one obtains $x_k = (k+1)\psi_t - k\psi_{t+1}$. Uniqueness follows from linearity. $\square$

**Theorem A.19** (Bias of the three strategies). *Let $M_r := \sup_{s \in [0,1]} \|\psi^{*(r)}(s)\|$ for $r = 1, 2, 3, 4$. Then, uniformly in $t$ and $j$:*

(A) *For two-point linear extrapolation,*

$$\|\mathrm{Bias}(\widehat{\psi}_{t-j}^A)\| \leq \tfrac{1}{2} j(j+1) M_2 \Delta^2 = O(j^2\Delta^2).$$

*Moreover, the exact Taylor expansion is*

$$\mathrm{Bias}(\widehat{\psi}_{t-j}^A) = -\tfrac{1}{2} j(j+1) \psi_t^{*\prime\prime}\Delta^2 + \tfrac{1}{6}(j^3 - j) \psi_t^{*(3)}\Delta^3 - \tfrac{1}{24}(j^4 + j) \psi_t^{*(4)}\Delta^4 + O(\Delta^5).$$

(B) *For interval reuse, letting $j = 2k$ or $j = 2k+1$,*

$$\|\text{Bias}(\widehat{\psi}_{t-2k}^B)\| \le 2k\, M_1\, \Delta + \tfrac{1}{2}(2k)^2 M_2 \Delta^2 + \tfrac{1}{6}(2k)^3 M_3 \Delta^3 + O(\Delta^4) = O(j\Delta),$$

$$\|\text{Bias}(\widehat{\psi}_{t-(2k+1)}^B)\| \le 2k\, M_1\, \Delta + (2k^2+2k+1)M_2\Delta^2 + O(\Delta^3) = O(j\Delta).$$

*The leading-order expansions are, respectively,*

$$\text{Bias}(\widehat{\psi}_{t-2k}^B) = 2k\,\psi_t^{*\prime}\Delta - 2k^2\,\tfrac{\psi_t^{*\prime\prime}}{1}\Delta^2 + O(\Delta^3),$$

$$\text{Bias}(\widehat{\psi}_{t-(2k+1)}^B) = 2k\,\psi_t^{*\prime}\Delta - (2k^2+2k+1)\psi_t^{*\prime\prime}\Delta^2 + O(\Delta^3).$$

(C) *For pure reuse,*

$$\|\text{Bias}(\widehat{\psi}_{t-j}^C)\| \le j\, M_1\, \Delta + \tfrac{1}{2}j^2 M_2 \Delta^2 + \tfrac{1}{6}j^3 M_3 \Delta^3 + O(\Delta^4) = O(j\Delta),$$

*with leading expansion* $\text{Bias}(\widehat{\psi}_{t-j}^C) = j\,\psi_t^{*\prime}\Delta - \tfrac{1}{2}j^2\psi_t^{*\prime\prime}\Delta^2 + O(\Delta^3).$

*Proof.* Fix a uniform grid with step size $\Delta > 0$ so that $s_{t+1} = s_t + \Delta$ and $s_{t-j} = s_t - j\Delta$. Let $\psi^* : [0,1] \to \mathbb{R}^d$ be $C^4$ and denote derivatives at $s_t$ by

$$\psi := \psi^*(s_t), \qquad \psi' := \psi^{*(1)}(s_t), \quad \psi'' := \psi^{*(2)}(s_t), \quad \psi''' := \psi^{*(3)}(s_t), \quad \psi'''' := \psi^{*(4)}(s_t).$$

The observations are $\psi_u = \psi^*(s_u) + \eta_u$ with $\mathbb{E}[\eta_u] = 0$. Hence for any affine estimator $\widehat{\psi}_{t-j}$ based on $(\psi_t, \psi_{t+1})$, its bias is

$$\text{Bias}(\widehat{\psi}_{t-j}) = \mathbb{E}[\widehat{\psi}_{t-j}] - \psi^*(s_{t-j}) = \widehat{\psi}_{t-j}^{\text{det}} - \psi^*(s_{t-j}), \tag{A.15}$$

where $\widehat{\psi}_{t-j}^{\text{det}}$ is obtained by replacing $(\psi_t, \psi_{t+1})$ with $(\psi^*(s_t), \psi^*(s_{t+1}))$.

Taylor expansion at $s_t$ up to order four gives

$$\psi^*(s_{t+1}) = \psi + \psi'\Delta + \tfrac{1}{2}\psi''\Delta^2 + \tfrac{1}{6}\psi'''\Delta^3 + \tfrac{1}{24}\psi''''\Delta^4 + O(\Delta^5), \tag{A.16}$$

$$\psi^*(s_{t-j}) = \psi - j\psi'\Delta + \tfrac{1}{2}j^2\psi''\Delta^2 - \tfrac{1}{6}j^3\psi'''\Delta^3 + \tfrac{1}{24}j^4\psi''''\Delta^4 + O(\Delta^5). \tag{A.17}$$

All $O(\cdot)$ terms are uniform in $t, j$ once bounded by

$$M_r := \sup_{s \in [0,1]} \|\psi^{*(r)}(s)\|, \qquad r = 1, 2, 3, 4.$$

**Strategy (A).** By definition,

$$\widehat{\psi}_{t-j}^{\text{det},A} = (j+1)\psi^*(s_t) - j\psi^*(s_{t+1}) = (j+1)\psi - j\big(\psi + \psi'\Delta + \tfrac{1}{2}\psi''\Delta^2 + \tfrac{1}{6}\psi'''\Delta^3 + \tfrac{1}{24}\psi''''\Delta^4\big) + O(\Delta^5).$$

Simplifying gives

$$\widehat{\psi}_{t-j}^{\text{det},A} = \psi - j\psi'\Delta - \tfrac{j}{2}\psi''\Delta^2 - \tfrac{j}{6}\psi'''\Delta^3 - \tfrac{j}{24}\psi''''\Delta^4 + O(\Delta^5). \tag{A.18}$$

Subtracting equation A.17 from equation A.18 yields

$$\text{Bias}(\widehat{\psi}_{t-j}^A) = -\tfrac{1}{2}j(j+1)\psi''\Delta^2 + \tfrac{1}{6}(j^3 - j)\psi'''\Delta^3 - \tfrac{1}{24}(j^4 + j)\psi''''\Delta^4 + O(\Delta^5),$$

and therefore

$$\|\text{Bias}(\widehat{\psi}_{t-j}^A)\| \le \tfrac{1}{2}j(j+1)M_2\Delta^2 + O(j^3\Delta^3) = O(j^2\Delta^2).$$

**Strategy (B).** For even $j = 2k$, one has $\widehat{\psi}_{t-2k}^{\text{det},B} = \psi$, hence

$$\text{Bias}(\widehat{\psi}_{t-2k}^B) = \psi - \psi^*(s_{t-2k}) = 2k\,\psi'\Delta - 2k^2\psi''\Delta^2 + \tfrac{4}{3}k^3\psi'''\Delta^3 + O(\Delta^4).$$

This implies

$$\|\text{Bias}(\widehat{\psi}_{t-2k}^B)\| \le 2kM_1\Delta + \tfrac{1}{2}(2k)^2 M_2\Delta^2 + \tfrac{1}{6}(2k)^3 M_3\Delta^3 + O(\Delta^4) = O(j\Delta).$$

For odd $j = 2k+1$, one has $\widehat{\psi}_{t-(2k+1)}^{\det,B} = 2\psi - \psi^*(s_{t+1})$. Expanding and subtracting equation A.17 with $j = 2k+1$ yields

$$\text{Bias}(\widehat{\psi}_{t-(2k+1)}^B) = 2k\,\psi'\Delta - (2k^2 + 2k + 1)\psi''\Delta^2 + O(\Delta^3),$$

so that

$$\|\text{Bias}(\widehat{\psi}_{t-(2k+1)}^B)\| \le 2kM_1\Delta + (2k^2 + 2k + 1)M_2\Delta^2 + O(\Delta^3) = O(j\Delta).$$

**Strategy (C).** Here $\widehat{\psi}_{t-j}^{\det,C} = \psi$, so

$$\text{Bias}(\widehat{\psi}_{t-j}^C) = \psi - \psi^*(s_{t-j}) = j\psi'\Delta - \tfrac{1}{2}j^2\psi''\Delta^2 + \tfrac{1}{6}j^3\psi'''\Delta^3 + O(\Delta^4),$$

and thus

$$\|\text{Bias}(\widehat{\psi}_{t-j}^C)\| \le jM_1\Delta + \tfrac{1}{2}j^2M_2\Delta^2 + \tfrac{1}{6}j^3M_3\Delta^3 + O(\Delta^4) = O(j\Delta).$$

Combining the three strategies completes the derivation of the bias bounds and their leading expansions. □

**Theorem A.20** (Variance growth)**.** *Under the noise model above and independence across grid points, the variances are:*

$$\text{Var}(\widehat{\psi}_{t-j}^A) = \big((j+1)^2 + j^2\big)\sigma^2 I_d, \qquad \text{Var}(\widehat{\psi}_{t-j}^B) = \begin{cases} \sigma^2 I_d, & j \text{ even,} \\ 5\sigma^2 I_d, & j \text{ odd,} \end{cases} \qquad \text{Var}(\widehat{\psi}_{t-j}^C) = \sigma^2 I_d.$$

*Proof.* Each estimator is a fixed linear combination $w_0\psi_t + w_1\psi_{t+1}$ with weights $(w_0, w_1) = ((j+1), -j)$ for (A), $(1,0)$ or $(2,-1)$ for (B), and $(1,0)$ for (C). With homoscedastic uncorrelated noise, $\text{Var}(w_0\psi_t + w_1\psi_{t+1}) = (w_0^2 + w_1^2)\sigma^2 I_d$. □

**Corollary A.21** (Dominant orders of MSE)**.** *Let $h := j\Delta$. As $\Delta \to 0$ with $j$ possibly growing, the mean-squared errors satisfy*

$$\text{MSE}(\widehat{\psi}_{t-j}^A) = \Theta(h^4) + \Theta(j^2\sigma^2), \qquad \text{MSE}(\widehat{\psi}_{t-j}^B) = \Theta(h^2) + \Theta(\sigma^2), \qquad \text{MSE}(\widehat{\psi}_{t-j}^C) = \Theta(h^2) + \Theta(\sigma^2).$$

*Proof.* Combine Theorems A.19 and A.20. The squared-bias for (A) is $\Theta((j^2\Delta^2)^2) = \Theta(h^4)$, while for (B) and (C) it is $\Theta((j\Delta)^2) = \Theta(h^2)$. The variance orders are given in Theorem A.20. □

*Remark* A.22 (Interpretation)*.* Strategy (A) achieves second-order bias $\Theta(h^2)$ but pays a variance that grows quadratically with the number of skipped steps. Strategies (B) and (C) maintain constant variance independent of $j$ but can only guarantee first-order bias $\Theta(h)$. In low-noise, short-jump regimes, (A) enjoys a strictly better MSE due to its $\Theta(h^4)$ squared bias. However, once the jump length exceeds one step or the noise level becomes higher, the variance term of (A) dominates, and (B)/(C) may become preferable. In summary, extrapolation tends to overshoot, while reuse remains stable but sacrifices precision.

