# OpenReview forum: "Accelerating Denoising Generative Models is as Easy as Predicting Second-Order Difference"
_ICLR.cc/2026/Conference — Submitted to ICLR 2026_

### Official Review · Reviewer_34k7 · 2025-10-29

**Soundness:** 2
**Presentation:** 2
**Contribution:** 2
**Rating:** 4
**Confidence:** 3

**Summary:**

ZEUS accelerates diffusion/flow sampling without retraining by leveraging a budget‑induced “signal scarcity,” where each three‑step window contains at most one fresh denoiser call. Using only the fresh output and its backward difference, it derives a uniquely minimal, affine‑exact second‑order predictor proven BLUE among linear two‑point estimators, and stabilizes chaining via an interleaved zig–zag schedule that avoids back‑to‑back extrapolations. The zero‑overhead plug‑in is backbone‑ and parameterization‑agnostic, requiring no caches or architectural changes, and consistently pushes the speed–fidelity Pareto frontier outward on images and video with substantial end‑to‑end speedups and improved perceptual similarity.

**Strengths:**

1. This paper conducts extensive experiments to show the superiority.
2. It provides a theoretical analysis.

**Weaknesses:**

1. This paper is hard to read due to poor writing.
2. The proposed method is very similar to Taylor ($\mathcal{O}=1$) for obtaining $\widehat\psi_{t-1}$.
3. I believe the method induces lots of memory costs, which is not justified by the authors.
4. The authors only provide comparisons with DiCache for FLUX. I believe they should include comparisons for the other models.  Also, the settings for baselines in this paper are not clear.
5. The performance of TaylorSeer is inconsistent (a huge gap) with its original paper (Tab. 1 for FLUX).

**Questions:**

N/A

---

> ### Author Response · Authors · 2025-12-03
> **Rebuttal for Reviewer 34k7**
>
> We appreciate the reviewers’ feedback and address the points below.
>
> **[W1]** We thank the reviewer for the feedback. If there are specific passages that read as unclear or poorly written, we would greatly appreciate concrete pointers and will revise them carefully.
>
> **[W2]** We agree that our predictor is mathematically equivalent to a first-order Taylor extrapolation, and we explicitly state this connection in the manuscript. Our contribution is not to propose a new heuristic Taylor predictor, but rather (i) to provide a theoretical explanation of **why** this extremely simple second-order predictor is consistently effective under aggressive acceleration, and (ii) to design a principled scheme that preserves accuracy and keeps the prediction variance bounded across a wide range of samplers and models. We believe this combination of analysis, scheme design, and extensive empirical validation goes significantly beyond merely re-using Taylor (O=1).
>
>
> **[W3]** We respectfully disagree with the claim that the extra memory cost is “formidable.” While ZEUS does introduce an additional cache for the intermediate latents $\mathcal O(C_{\text{latent}} \times H' \times W')$, this does **not** alter the asymptotic memory complexity, which remains at $\mathcal{O}(M_w + M_{\text{act}})$. Empirically, our experiments confirm that the actual memory overhead is negligible in practice. For example, in the Flux v1 test, the peak memory usage increased by just **0.01 GB** (from 33.83 GB to 33.84 GB) with ZEUS enabled. Similarly, on Wan 2.1 14b T2V, the overhead was only **0.02 GB** (46.04 GB vs. 46.06 GB). The extra cache required at most **15 MB**--orders of magnitude smaller than the memory cost of model activations (in GiB). Therefore, both the theoretical and empirical evidence demonstrate that ZEUS's additional memory overhead is minimal.
>
> **[W4]** We appreciate the concern about DiCache coverage and baseline settings. DiCache is only available for certain models (currently FLUX image generation and specific video models such as Wan and Hunyuan), so our comparison with DiCache is necessarily limited to these supported backbones. For other image models used in our experiments (e.g., SD-*), DiCache does not provide integration. Meanwhile, ZEUS outperforms DiCache in perceptual similarity in an identical speedup ratio.
>
> For all baselines, we rely on the official Diffusers implementations. Experimental ettings follow the original papers unless otherwise stated. We will explicitly clarify (i) for which models DiCache is available and (ii) the shared configuration used for all baselines in the revision.
>
>
>
> **[W5]** We believe that faithfulness between accelerated samples and the unmodified baseline is the main objective of training-free acceleration. Following prior work such as SADA and SVG2, we therefore evaluate perceptual similarity (LPIPS) and distributional similarity (FID), which directly measure how closely the accelerated sampler matches the baseline outputs. Methods like TaylorSeer and ToCa also report metrics related to text–image alignment or aesthetics (e.g., CLIP-based or ImageReward scores), which are complementary but not the primary focus of our evaluation protocol.
> Meanwhile, our experiment results on TaylorSeer matches DiCache's experiment results on TaylorSeer.

---

### Official Review · Reviewer_4EiC · 2025-10-30

**Soundness:** 2
**Presentation:** 2
**Contribution:** 2
**Rating:** 4
**Confidence:** 3

**Summary:**

This paper introduces ZEUS (Zero-cost Extrapolation-based Unified Sparsity), a training-free and backbone-agnostic method for accelerating denoising generative models—such as diffusion and flow-based architectures—without modifying model weights, retraining, or using feature caches. Empirically, ZEUS is evaluated across five backbones (e.g., Stable Diffusion 2, SDXL, Flux.1-dev, Wan 2.1, CogVideoX v1.5) and two solvers (Euler and DPM++), consistently pushing the speed–fidelity Pareto frontier forward. It achieves up to 3.2× end-to-end speedup while maintaining or improving perceptual metrics such as LPIPS, FID, and PSNR.

**Strengths:**

1. The authors formally prove the BLUE optimality and second-order accuracy of the proposed predictor, and justify why higher-order extrapolants are suboptimal or unstable under limited fresh computation. The combination of theory (bias–variance characterization, affine invariance) with practical implementation (zig–zag reuse schedule) demonstrates high technical soundness.

2. ZEUS has strong practical and scientific significance. It provides a zero-cost, architecture-agnostic plug-in that can be directly applied to large-scale diffusion and flow models such as SDXL, Flux, and Wan2.1. The method delivers up to 3.2× inference acceleration while maintaining or even improving fidelity (LPIPS/FID), which represents a meaningful advance in the efficiency of generative modeling.

**Weaknesses:**

1. Evaluation scope and metric coverage.

Although the experiments are extensive, the paper primarily reports traditional similarity metrics (PSNR, LPIPS, FID) and qualitative visual comparisons. However, these metrics capture perceptual closeness rather than semantic or compositional fidelity. For text-to-image generation, including evaluations such as GenEval or DPG-Bench would better reflect whether ZEUS preserves prompt alignment and fine-grained attribute consistency after aggressive skipping. This would strengthen claims of maintaining “fidelity” beyond pixel-level similarity.

2. No discussion on potential integration with caching or mixed-precision methods.

While ZEUS is intentionally designed as a zero-overhead plug-in, the paper stops short of exploring synergy with complementary acceleration families, such as feature-cache re-use (DeepCache, AB-Cache). Discussing how ZEUS could combine with these orthogonal techniques to achieve compound acceleration would expand its utility.

**Questions:**

Lack of analysis on strong modern backbones.

All reported results focus on mid-scale or publicly available diffusion/flow models (e.g., SDXL, Flux.1-dev, Wan2.1). The paper does not evaluate on more recent state-of-the-art text-to-image backbones like FLUX-Krea, and Qwen-Image that exhibit stronger coupling between text and visual features. Since ZEUS claims architecture-agnostic generalization, demonstrating performance on these modern DiT-based models would further substantiate scalability and practical relevance.

---

> ### Author Response · Authors · 2025-12-03
> **Rebuttal for Reviewer 4EiC**
>
> We thank the reviewers for the constructive suggestions.
>
> **W1.** We thank the reviewer for raising the issue of semantic and compositional evaluation. Benchmarks like GenEval and DPG-Bench were originally designed to assess absolute text–image alignment of fully trained models. In this work, our primary goal is to assess how much a **fixed, pretrained diffusion model** changes under acceleration, so we follow a standard evaluation protocol used in recent training-free acceleration strategies such as AdaptiveDiffusion [1], SADA [2], and SVG2 [3], which rely on PSNR, LPIPS, FID to allow direct comparison on generation faithfulness.
>
>
> **W2.** We appreciate the reviewer’s suggestion to discuss integration with caching and mixed-precision techniques. Our goal in designing ZEUS was to provide a zero-overhead, training-free plug-in with O(1) additional memory, which makes it particularly attractive in memory-constrained settings. In contrast, feature-caching methods such as TaylorSeer require storing intermediate feature maps, incurring nontrivial memory overhead.
>
> ZEUS applies a uniform reduction with a zig-zag reuse pattern, so under aggressive acceleration full denoiser evaluations are relatively sparse and the last fully evaluated feature is **no longer a good local approximation for the current step**. In this regime, naively adding feature caching (e.g., DeepCache, TaylorSeer) on top of ZEUS yields little extra benefit and can even harm perceptual quality, while also discarding the O(1) memory advantage of ZEUS.
>
>
>
> Mixed-precision acceleration is largely orthogonal to our contribution. Since ZEUS only decides "when the denoiser is called", we expect it to be compatible with mixed-precision implementations of the backbone denoiser. A systematic joint study is left for future work. We will clarify these points and add a dedicated discussion on potential integrations with caching and mixed-precision methods in the revised manuscript.
>
>
>
> **Q1.** We now apply ZEUS on the most recent, and strongest 64B flux-2-Dev. The result is as follow, demonstrating ZEUS's **effectiveness and easily adaption** on state-of-the-art models without extra optimization.
>
>
> **Table 1: FLUX-2-dev (64B, 4-bit quantization)**
> | Metric                                    | ZEUS (Medium) | ZEUS (Turbo) |
> |-------------------------------------------|--------------:|-------------:|
> | PSNR ↑                                    |        31.593 |       23.115 |
> | LPIPS ↓                                   |         0.029 |        0.114 |
> | SSIM ↑                                    |         0.938 |        0.811 |
> | Peak Memory (4-bit quantization)      |      23370 MB |     23370 MB |
> | Speedup Ratio ↑                           |          2.07 |         3.34 |
>
>
>
> Refs:
>
> [1] Ye, Hancheng, et al. "Training-free adaptive diffusion with bounded difference approximation strategy." Advances in Neural Information Processing Systems 37 (2024): 306-332. https://proceedings.neurips.cc/paper_files/paper/2024/file/00d1f03b87a401b1c7957e0cc785d0bc-Paper-Conference.pdf
>
> [2] Jiang, Ting, et al. “SADA: Stability-guided Adaptive Diffusion Acceleration.” In Proceedings of the 42nd International Conference on Machine Learning (ICML 2025), PMLR 267: 27649–27669, 2025. https://openreview.net/forum?id=ThMQfsBnje
>
> [3] Yang, Shuo, et al. “Sparse VideoGen2: Accelerate Video Generation with Sparse Attention via Semantic-Aware Permutation.” In Advances in Neural Information Processing Systems 39 (NeurIPS 2025), Spotlight, 2025. https://openreview.net/forum?id=WPU17d1l7R

---

### Official Review · Reviewer_enNh · 2025-10-31

**Soundness:** 3
**Presentation:** 3
**Contribution:** 2
**Rating:** 4
**Confidence:** 3

**Summary:**

This paper proposes a training-free acceleration method for diffusion models named ZEUS. Based on an information scarcity assumption, the authors derive that a second-order difference predictor is the only form satisfying the BLUE condition. They further introduce an alternating zig–zag reuse mechanism to balance stability and accuracy. The method is zero-cost, requires no feature cache or structural modification, and can be directly embedded into any diffusion or flow-based model during inference. Experiments on Stable Diffusion, SDXL, Flux, Wan, and CogVideoX demonstrate consistent acceleration while maintaining or even improving generation quality, showing that ZEUS is a simple, stable, and general acceleration paradigm.

**Strengths:**

1. **Theoretical simplicity with solid foundation.** The paper rigorously derives that, under specific conditions, the second-order difference serves as the only optimal estimator satisfying the BLUE criterion, leading to a training-free and architecture-invariant extrapolation scheme.

2. **Efficient and stable design.** The proposed Zig–Zag reuse mechanism effectively suppresses multi-step extrapolation drift while maintaining the accuracy of second-order prediction, achieving stable acceleration with zero additional computational cost.

3. **Comprehensive experiments and strong generality.** ZEUS consistently improves the speed–quality trade-off across multiple image and video diffusion models, verifying its practicality and plug-and-play generalization capability.

**Weaknesses:**

1. **Overly structured core assumption.** The key theoretical derivation relies on a constrained premise—when aiming for acceleration beyond 2×, only one of every three consecutive steps can involve a fresh denoiser call. In real-world inference, non-uniform or adaptive scheduling is common, where this assumption may not hold. The paper has not provided theoretical or empirical evidence that the optimality can generalize to such practical cases.

2. **Lack of global stability and error analysis.** While the paper discusses overshoot phenomena and gives local bias–variance scaling laws, these analyses are confined to fixed short segments. In actual diffusion sampling, the process is a long chained integration. The paper does not provide a global upper bound or long-term error propagation analysis, making current stability conclusions largely empirical.

3. **Limited comparison and ablation scope.** The experiments mainly compare with training-free baselines, which are not strictly comparable to ZEUS. Stronger baselines or hybrid cache–predict strategies could be added. Moreover, the ablation design focuses primarily on the authors’ own setting, limiting the generality of the conclusions.

**Questions:**

1. **Theory.** Under non-uniform or adaptive time-step scheduling, does the theoretical optimality of the second-order predictor still hold?
Can the authors provide a global stability or error bound for the zig–zag reuse process across long sampling trajectories?
If a formal bound is not feasible, what is the empirically observed maximum jump length before instability occurs?

2. **Experiments.** It would strengthen the paper to include stronger baselines and high-order solvers for comparison. In addition, although the paper repeatedly claims model-agnostic generalization, experiments are mostly in visual diffusion models; evaluations on audio or text diffusion tasks (with different smoothness characteristics) are needed to validate the generality claim.

3. **Implementation cost.** Although the method is described as zero-cost at the operator level, additional tensor construction, copying, and logic overhead may still exist. Could the authors provide quantitative profiling data to verify that the reported acceleration indeed reflects the end-to-end wall-clock gain?

---

> ### Author Response · Authors · 2025-12-03
> **Rebuttal for Reviewer enNh**
>
> We thank the reviewers for their thoughtful comments.
>
> **[W1]** We thank the reviewer for raising this concern. The “one fresh evaluation in every three steps” pattern arises from our theoretical analysis under a simple uniform reduction rule; it is not a hard constraint of ZEUS. In practice, ZEUS can be applied with non‑uniform or even random patterns.
>
> To verify this, we ran an additional experiment on FLUX.1‑dev with an Euler solver, where we **randomly reduce 50% steps between step 10 to 45** and apply ZEUS **without langrange interpolation**. Random 50% reduction achieves an LPIPS of **0.0581** over 200 samples with a generation time of 2909.78s, compared to 4817.9s for the baseline, corresponding to a **1.65$\times$** speedup without degrading sample quality. Similarly, with Wan2.1-T2V-14B-Diffusers model on VBench, we achieves **1.55$\times$** speedup with **0.0294 LPIPS**. These results provide empirical evidence that the acceleration and stability of our method extend beyond the structured 1-in-3 design. That said, we emphasize that the uniform skip scheme is not only the analytically cleanest setting but also reflects our design philosophy. We do not recommend arbitrarily extreme or excessively long skip spans, as such configurations may cause error explosion (see Appendix A.5). In contrast, prior adaptive-scheduling approaches such as [1, 2] employ non-uniform skips not for theoretical optimality, but because their adaptive mechanisms prevent them from achieving a simple and elegant uniform pattern. We will clarify these insights and the recommended usage patterns in the revised paper.
>
>
> **[W2]** We thank the reviewer for this thoughtful comment on stability. Intuitively, our multi-step bounds in Appendix A.5 control the MSE within each skipped segment uniformly over time and skip length. Our experiment has supported our claim and show solid evidence on controlled error even under a turbo level skipping.
>
>
> **[W3]** We thank the reviewer for raising this point regarding the comparison scope. We respectfully disagree with the statement that our comparisons with training-free methods only are unfair. In fact, **ZEUS is itself a training-free method**, so comparing it against other training-free approaches under identical setting and standard evaluation suite is appropriate. To the best of our knowledge, we have included all recent popular and strong open-source training-free baselines, including hybrid cache-predict approachs (i.e., TaylorSeer, ToCa), pure caching baselines (i.e., DeepCache, TeaCache), and adaptive step-allocation methods (i.e., AdaptiveDiffusion, SADA).
>
> It is worth mentioning that another hybrid cache-predict approachs, AB-Cache, has not publically released code, so we are currently unable to include it as a fair quantitative baseline.
>
>
>
>
>
> Refs:
>
> [1] Ye, Hancheng, et al. "Training-free adaptive diffusion with bounded difference approximation strategy." Advances in Neural Information Processing Systems 37 (2024): 306-332. https://proceedings.neurips.cc/paper_files/paper/2024/file/00d1f03b87a401b1c7957e0cc785d0bc-Paper-Conference.pdf
>
> [2] Jiang, Ting, et al. “SADA: Stability-guided Adaptive Diffusion Acceleration.” In Proceedings of the 42nd International Conference on Machine Learning (ICML 2025), PMLR 267: 27649–27669, 2025. https://openreview.net/forum?id=ThMQfsBnje

---

> ### Author Response · Authors · 2025-12-03
> **Rebuttal for Reviewer enNh (Questions)**
>
> **[Q1]** (see Weakness 1,2)
>
> **[Q2-1]** We appreciate the suggestion and agree that comparing against strong high-order solvers is important. Our method is inherently solver-agnostic: ZEUS operates on denoiser outputs and can be combined with numerical solvers of any order. In the current draft we already evaluate ZEUS with Euler (first order) and DPM-Solver++ (second order), which are standard, widely used samplers for diffusion models, in addition to a broad suite of recent training-free acceleration paradigms.
>
> **Table 2: Ablation on higher order solver**
>
> | Solver   | PSNR  | LPIPS ↓ | FID ↓ | Speedup Ratio ↑ |
> |---------|-------|---------|-------|------------------|
> | Euler-1 | 28.66 | 0.095   | 3.87  | 1.85×            |
> | DPM++-2 | **29.17** | 0.084   | 3.59  | 1.87×            |
> | DPM++-3 | 29.05 | **0.079** | **3.28** | **1.88×**        |
>
> We have evaluated ZEUS-Medium with DPM-Solver++ (3rd order) as shown above. Moving from a 1st- to a 3rd-order solver yields **strictly improved perceptual and distributional fidelity (LPIPS/FID)** while maintaining the PSNR and speedup. This ablation provides strong empirical support for our theoretical claim that a second-order difference predictor is the minimal yet sufficient building block: it remains effective across 1st-, 2nd-, and even 3rd-order solvers and **continues to unlock higher fidelity as the underlying solver order increases**.
>
> **[Q2-2]** Our “model-agnostic” claim is intended within the class of continuous-time diffusion models whose sampling procedure is a numerical integration of an ODE/SDE over time with Gaussian noise (e.g., standard image / video diffusion). ZEUS explicitly relies on the smooth denoiser trajectory along this continuous-time path and on the fact that the sampler exposes a small number of ODE/SDE steps.
>
> Current text diffusion models, however, are typically discrete token models with masked-language-model-style objectives and bidirectional encoders, and do not conform to this continuous-time ODE/SDE framework. Applying ZEUS in that setting would likely require non-trivial modifications and a different theoretical treatment, and we therefore do not claim that our current analysis directly covers these architectures.
>
>
> **[Q3]** Thank you for raising this important point. We agree that “zero-cost at the operator level” must be evaluated in terms of *end-to-end wall-clock performance*, and we clarify both our implementation and measurement methodology.
>
> First, ZEUS operates only on **latent-level tensors** (model outputs), not internal feature maps. At each timestep, we cache at most $K \leq 3$ recent denoiser outputs $\psi_t$ of shape $(B, c_{\text{latent}}, H', W')$ (and, for video, an extra temporal dimension). Thus, the additional per-step buffer is bounded by $K \cdot c_{\text{latent}} \cdot H' \cdot W'$ elements.
>
> These buffers are **negligible** compared to the multi-GB memory requirements for backbone activations (e.g., self-attention and FFN layers), even with optimized kernels (e.g., FlashAttention). In asymptotic terms, ZEUS adds only a small, constant-factor increase in latent-sized tensors and simple linear ops, while the dominant cost remains the U-Net/DiT forward pass.
>
> Second, all reported speedups (Tables 1-3) are **end-to-end wall-clock times**-from the first denoiser call to the final decoded output—using identical hardware and mixed-precision settings for both baseline and ZEUS. Any additional overhead from tensor operations or control logic is therefore already included in our acceleration numbers.
>
> We will clarify these points in the revised manuscript by (i) explicitly describing the cached state and its size, and (ii) adding a profiling table in the appendix that separates backbone compute from ZEUS-specific overhead, making the end-to-end nature of our reported acceleration fully transparent.

---

### Meta-Review · Area_Chair_SZNW · 2026-01-03

**Summary:**

The reviewers unanimously rated the paper as marginally below the acceptance threshold (Score: 4).
Specifically,
 Reviewer enNh questioned the validity of the core theoretical assumption regarding uniform scheduling in practical scenarios and highlighted the absence of a global stability analysis or error bounds for the proposed Zig-Zag reuse mechanism.

 Concerns about evaluation comprehensiveness were raised by Reviewer 4EIC, who criticized the reliance on perceptual metrics (LPIPS, FID) without semantic alignment benchmarks (e.g., GenEval) and noted the lack of analysis on SOTA backbones or integration with caching methods.

Furthermore, Reviewer 34k7 challenged the method's novelty by noting its similarity to first-order Taylor expansion and raised doubts regarding potential memory costs and the fairness of baseline comparisons, specifically noting inconsistencies with TaylorSeer results.

**Reviewer Concerns:**

The rebuttal successfully addressed some of the empirical and implementation concerns, specifically regarding computational costs and memory overhead by clarifying that reported speedups are end-to-end wall-clock times with negligible memory increase . The authors also satisfied Reviewer 4EIC's request for testing on state-of-the-art architectures by adding results for the 64B FLUX-2-dev model , and effectively refuted Reviewer 34k7's critique regarding the method's similarity to Taylor expansion by emphasizing the theoretical optimality (BLUE) and the Zig-Zag scheme as key contributions . However, regarding Reviewer enNh's concern on non-uniform/adaptive scheduling , while the authors provided empirical evidence using random skipping , this response arguably conflates random skipping with true non-uniform step size theory, potentially masking the theoretical defect rather than theoretically justifying the method's optimality in adaptive settings. Furthermore, Reviewer enNh's request for a formal global error bound remains technically outstanding , and Reviewer 4EIC's concern regarding the lack of semantic alignment benchmarks (e.g., GenEval) was defended against rather than implemented , leaving the evaluation scope potentially narrow in that regard.

**Reviewer Scores:**

Reviewer 4EIC would likely keep the score or increase the score to 5. The reviewer's primary concerns were the lack of evaluation on state-of-the-art (SOTA) backbones and the absence of integration discussions with other methods . The authors directly addressed the SOTA concern by providing impressive results on the FLUX-2-dev (64B) model . They also provided a reasonable, albeit defensive, explanation for why integration with caching is counterproductive, while the refusal to add semantic benchmarks (GenEval) remains a sticking point.

Reviewer enNh would likely keep the score or increase the score to 5. This reviewer questioned the rigidity of the core assumptions, specifically regarding "Non-uniform/Adaptive Scheduling" . While the authors attempted to address this by running a "randomly skip 50% steps" experiment to claim generality , this response arguably conflates "random skipping" with true "non-uniform step size" theory, potentially masking the theoretical defect rather than resolving it. By substituting an empirical random test for the requested theoretical justification of adaptive scheduling, the authors may have failed to fully satisfy the reviewer's demand for rigor. Coupled with the continued absence of a formal global error bound , the reviewer might view the rebuttal as evasive regarding the theoretical flaws, limiting score increase.


Reviewer 34k7 would likely keep the score.  This reviewer was the most critical regarding novelty (calling it just Taylor expansion) and memory costs . The authors effectively debunked the memory cost concern with concrete data (0.01 GB increase). They also accepted the mathematical equivalence to Taylor expansion but successfully reframed the contribution around the optimality proof and the Zig-Zag strategy .

---

### Decision · Program_Chairs · 2026-01-26

Reject